# Quad-Polarimetric Multi-Scale Analysis of Icebergs in ALOS-2 SAR Data: A Comparison between Icebergs in West and East Greenland

**Johnson Bailey *** and **Armando Marino**

Faculty of Natural Sciences, The University of Stirling, Stirling FK9 4LA, UK; armando.marino@stir.ac.uk
* Correspondence: johnson.bailey@stir.ac.uk

**Abstract:** Icebergs are ocean hazards which require extensive monitoring. Synthetic Aperture Radar (SAR) satellites can help with this, however, SAR backscattering is strongly influenced by the properties of icebergs, together with meteorological and environmental conditions. In this work, we used five images of quad-pol ALOS-2/PALSAR-2 SAR data to analyse 1332 icebergs in five locations in west and east Greenland. We investigate the backscatter and polarimetric behaviour, by using several observables and decompositions such as the Cloude–Pottier eigenvalue/eigenvector and Yamaguchi model-based decompositions. Our results show that those icebergs can contain a variety of scattering mechanisms at L-band. However, the most common scattering mechanism for icebergs is surface scattering, with the second most dominant volume scattering (or more generally, clouds of dipoles). In some cases, we observed a double bounce dominance, but this is not as common. Interestingly, we identified that different locations (e.g., glaciers) produce icebergs with different polarimetric characteristics. We also performed a multi-scale analysis using boxcar 5 × 5 and 11 × 11 window sizes and this revealed that depending on locations (and therefore, characteristics) icebergs can be a collection of strong scatterers that are packed in a denser or less dense way. This gives hope for using quad-pol polarimetry to provide some iceberg classifications in the future.

**Keywords:** SAR; polarimetry; icebergs; Greenland; backscatter

## 1. Introduction

Icebergs are pieces of freshwater ice calved from glaciers or ice shelves. They are natural ocean hazards to ships and maritime activities [1,2]. Climate change is increasing the formation of new icebergs due to rapid melting and speeding up of glacial ice [3,4]. Remote sensing techniques, such as Synthetic Aperture Radar (SAR), have routinely been used to detect and analyse large icebergs [5,6], however, the understanding of the scattering mechanisms for iceberg backscattering is still a topic of research [7]. It is known that physical properties of icebergs may induce different backscattering behaviours [8].

Greenland and the Arctic are currently warming rapidly due to climate change [9]. Many of Greenland's glaciers are melting and icebergs are calving into the ocean much more rapidly [10]. These icebergs can drift into the Atlantic and into important shipping lanes. For this reason, monitoring the iceberg behaviour is critical to ensure the safety of maritime activities. Icebergs in the Arctic tend to be smaller and of irregular shape compared to their Antarctic counterparts, which are usually tabular in shape [11–13]. Arctic iceberg sizes can be relatively small with a few tens of metres outside the water. This presents a challenge for monitoring, as detection of smaller sized icebergs under 100 m in length is difficult [14]. Studying the scattering behaviour of icebergs is therefore important, if we need to identify smaller icebergs via PolSAR analysis. This knowledge will be important in the future when an

ad hoc PolSAR detector will be designed and it also opens the prospect to try to use the information derived on iceberg structure to improve iceberg classification

### 1.1. Icebergs in SAR

Icebergs are large chunks of freshwater ice which have calved from a glacier tongue, ice sheet or ice shelf [15]. They differ from sea ice, which forms from the surface of seawater itself, and so is made of saltwater. They can be classified based on their shape and size [16]. No single iceberg is of the exact same shape. Because ice can form at cold temperatures and melt at warm temperatures, the shape and structure of the iceberg is always changing. Classification of icebergs by shape and size is available [12–16]. Icebergs can be of either a tabular, wedge, pinnacle, drydock, or blocky shape [17]. Tabular icebergs have a flat surface with steep drops, wedge icebergs form a pyramid shape towards the top, pinnacle icebergs tend to have one or more spires, a drydock is a U-shaped berg that has been eroded, and blocky icebergs are similar to tabular shape bergs, but take more of a cubed shape. Sizes range from growler (10 m), bergy bit (5–15 m), small berg (15–60 m), medium berg (61–120 m), large berg (121–200 m) and very large berg (>200 m) [15].

Satellites with SARs on board such as ERS-1, ERS-2, Envisat and Sentinel-1 have a high spatial resolution of around 30 m, but this varies depending on the acquisition mode. This allows for the detection of smaller icebergs with an edge length of approximately 100 m [14]. Recent work has also been carried out using COSMO-SkyMed X-band imagery [18,19]. For classification purposes, SAR can detect from the higher range of medium bergs. Detection of icebergs smaller in size is, therefore, more complicated and problematic [20]. Although there are studies proposing backscattering models of icebergs, it is not clear how physical (e.g., shape, structure) and environmental (e.g., temperature, surface liquid water) conditions of icebergs can modify these backscattering models.

Icebergs are 3D objects and it is well known that the surface/terrain slope has a strong effect on SAR images [21], showing brightness modulations. In the case of icebergs, the geometric properties of icebergs would influence the image created by SAR, presenting bright layover areas and dark shadows.

Backscatter from icebergs can be rather similar to the clutter background when sea ice surrounds the iceberg. Indeed, several papers have outlined the difficulty of detecting icebergs embedded within sea ice [20]. When sea ice has an irregular or rough shape, such as the presence of ridges or hammocks, the backscattering from sea ice can become very bright and resemble the one from icebergs producing several false alarms [14].

Iceberg surface roughness plays a very important role in SAR detection, especially when the iceberg is melting. Additionally, most icebergs have internal structures with cracks or crevasses [22]. Icebergs more irregular in shape may be harder to detect [23]. Similarly, open water backscatter intensities sometimes have a stronger backscattering than small icebergs [16]. Iceberg radar backscattering coefficients are the sum of surface, multiple reflections and volume contributions of an iceberg. Backscattering intensity depends on iceberg shape, surface roughness and ice volume (fraction, shape, size of cracks, air bubbles and impurities) [24]. Smaller iceberg sizes may also be a result of image speckle lowering the spatial resolution. As a result, targets in single pixels are much harder to detect than those in multiple pixels [14].

The application of quad-pol SAR data can be used to increase our ability to distinguish between icebergs and sea ice [23]. This is due to the capability to identify multiple scattering mechanisms and avoid focusing on a fixed scattering mechanism, where the iceberg may be very weak or the background may be very strong.

Attempts to detect smaller icebergs are evident in the literature. Some studies show that by using SAR images at 30 m spatial resolution and adding polarimetric information, we can detect icebergs of approximately 100 m diameter and up [24]. Algorithms have been developed using SAR polarimetry and perturbation analysis enhancing the detection of ships and icebergs [25]. More recently, work was completed on a new dual-polarisation radio anomaly detector (iDPolRad) algorithm to detect smaller

icebergs [26]. It is established that richer information could be obtained if the use of quad-pol data was implemented [20,27].

The large-scale roughness and topographical features of icebergs are likely to influence the scattering behaviour. One study by Zakharov [6] summarised how multi-resolution and frequencies of SAR can lead to a more coherent knowledge of iceberg topography. Additionally, in one study from Herdes [28], interferometric SAR data were used to reconstruct icebergs topography. This shows that the heights of shadows can be used to analyse surface topography. Generally, darker shadows indicate more of a rough topography [28].

Similar studies have been done to estimate heights of icebergs. It is well known that an iceberg with a lower optical reflectance may indicate a higher backscatter on a SAR image, particularly in areas and patches of higher temperatures. One paper from Viehoff employed the use of Advanced Very High Resolution Radiometer (AVHRR) and used ERS-1 SAR data to identify areas of high backscatter on icebergs [29]. The shadows created by tabular icebergs were then used to estimate heights and corresponding draughts of the icebergs. However, this study contained no information regarding smaller icebergs, which may not show features such as shadows.

Finally, work has been carried out with multiple pixels [30], in which iceberg edges were detected. The edges of icebergs may prove critical in a SAR multi-scale analysis because a higher resolution or window size may introduce several scatterers to the pixels.

In Greenland, snow is prone to melting and rainwater may fall from clouds within areas of warmer water where the iceberg may drift. Additionally, when an iceberg topples over, it may show the bottom part which is covered in high-dielectric constant saline ice [31]. The wetness of the iceberg surface affects the backscattering. One study by Xiang [32] documented the time delayed reflections of L-band SAR in icebergs and found that surface wetness may decrease the ability for SAR pulses to reflect back from an iceberg.

Tabular icebergs that have been freshly calved also appear brighter, indicating that any surface wetness is a result of later events, such as snow melt or other precipitation [30]. Icebergs that stand out as dark targets are a result of reduced volume scattering and radiation being reflected specularly on top of wet surfaces [16].

Backscatter intensities have been recorded on icebergs in the Weddell Sea using ERS-1 C-band SAR. It has been documented that they have an intensity range of −6 to −4 dB [15]. The surrounding clutter may produce an intensity of less than −10 dB [15]. However, these figures were dependent on seasonal variation. An image taken in the summer is likely to show a different scattering intensity to an image taken in winter. In this work, we use SAR images taken in both summer and winter.

Generally, icebergs which are smaller in size and embedded within sea ice, prove difficult for SAR detection. Accurate iceberg identification is dependent on the ability to distinguish them from open water, sea ice and other polar targets, all under a range of interchangeable environmental and meteorological conditions. Properties of icebergs along with their surrounding conditions determine the brightness and reflectivity when interpreted on a SAR image. Significantly, paucity of data surrounding backscatter behaviour is limited. In this work, we apply a range of polarimetric parameters to icebergs shown in SAR images to try to address this limitation.

### 1.2. Aims and Objectives

The aim of this paper is to analyse backscatter and polarimetric behaviour based on a series of polarimetric parameters. These include the Cloude–Pottier decomposition [33], the Yamaguchi decomposition, Pauli RGB and an overall intensity operator (span). These PolSAR methodologies can shade some light on the understanding of the different scattering mechanisms contributing to the iceberg backscattering. Our hypothesis is that icebergs exhibit a combination of different scattering mechanisms. It is hoped that this will lead to the improvement of future iceberg classification.

## 2. Materials and Methods

Each SAR image was taken from the PALSAR-2 instrument aboard the ALOS-2 radar satellite over east, north and west Greenland. These data were collected under an open JAXA Announcement of Opportunity. A total of five images were selected for analysis, processed via calibration, construction of a covariance matrix, boxcar filtering and finally, PolSAR parameters. We then performed a multi-scale analysis using two window sizes. In this section, we specify the materials and methods carried out for the analysis. A description of the SAR data is presented in Table 1.

**Table 1.** ALOS-2/PALSAR-2 JAXA properties. Centre DMS coordinates are selected for latitude and longitude. Incidence angle range is min, centre and max. Note the ground resolution is for ALOS-2/PALSAR-2 quad-pol mode. Time is UTC.

| Image ID | Location | Lat/Lon (DMS) | Resolution | Incidence Angle Range (°) | Date/Time |
|---|---|---|---|---|---|
| ALOS2066231360-150815 | Blosseville Coast N | 68°02′13.2″N 30°19′58.8″W | 4.3 × 5.1 | 37, 39, 41.5 | 15/08/2015 01:26 |
| ALOS2064761430-150805 | Nuugaatsiaq | 71°25′26.4″N 53°26′52.8″W | 4.3 × 5.1 | 37, 39, 41.5 | 05/08/2015 02:48 |
| ALOS2064461300-150803 | Isortoq | 65°07′08.4″N 39°13′37.2″W | 4.3 × 5.1 | 37, 39, 41.5 | 03/08/2015 02:07 |
| ALOS2057951350-150620 | Blosseville Coast S | 67°19′1.2″N 32°37′33.6″W | 4.3 × 5.1 | 29, 31, 33.6 | 20/06/2015 01:26 |
| ALOS2191031530-171206 | Savissivik | 75°52′19.2″N 62°10′48″W | 4.3 × 5.1 | 29, 31, 33.6 | 06/12/2017 02:52 |

### 2.1. SAR Processing and Iceberg Detection

In this section, we outline the procedure carried out to process the SAR dataset in order to evaluate the polarimetric information from the icebergs. A summary of steps is presented in Figure 1.

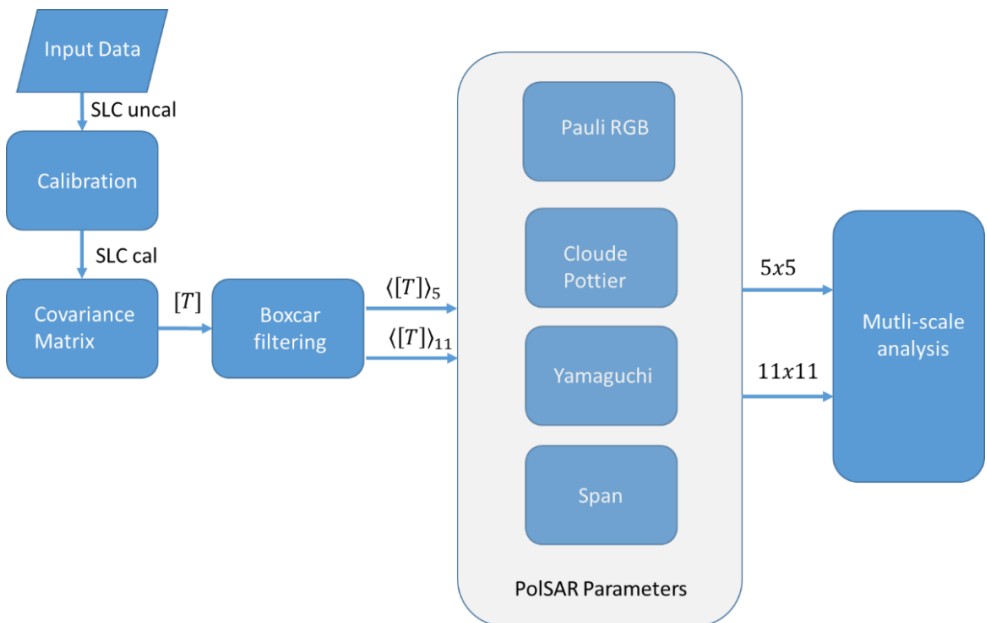

**Figure 1.** Block diagram outlining the methods of the study. The covariance matrix was built before main multi-scale analysis. Note that Pauli RGB refers to the RGB image and without it, the other steps would not have been carried out.

The first step was to input ALOS-2 quad-pol data and calibrate them into the appropriate SAR real and imaginary parts for each image. The radiometric calibration we applied on all images simply

removes the output scaling applied by the SAR focusing processor and formatting as sigma naught (normalised radar cross section). However, it does not convert data in dB, since we are interested in the Single Look Complex (SLC) format. The next step in processing the quad-pol data was to derive the structure of the covariance matrix, which will reveal the second order statistics of the partial target once the averaging is applied. We applied averaging using two filters either of $5 \times 5$ (which corresponds to $25.5 \times 21.5$ m) or $11 \times 11$ (which corresponds to $56.1 \times 47.3$ m). After the filtering, the data (in covariance matrix format) are ready to be processed for extracting decomposition parameters (Cloude–Pottier, Yamaguchi) or other observables (Pauli RGB and span). A list of all the polarimetric parameters used is in Table 2.

**Table 2.** Polarimetric parameters. Note the Yamaguchi parameters had their orientation removed and the span is a separate parameter that was deduced independent of a decomposition.

| Parameter | Type | Notes |
| --- | --- | --- |
| Alpha | Cloude–Pottier Decomposition | - |
| Entropy | Cloude–Pottier Decomposition | - |
| Beta | Cloude–Pottier Decomposition | - |
| Anisotropy | Cloude–Pottier Decomposition | - |
| Span | Observable | - |
| Y Double | Yamaguchi Decomposition | Orientation removed |
| Y Helix | Yamaguchi Decomposition | Orientation removed |
| Y Surface | Yamaguchi Decomposition | Orientation removed |
| Y Volume | Yamaguchi Decomposition | Orientation removed |

All these parameters are used to evaluate the polarimetric and brightness behaviour of icebergs. Please note, some of the parameters such as the Yamaguchi components are expressed in dB to ease the visualisation. The dB was applied to the outcome of the Yamaguchi.

The final step used here is the identification of icebergs, which feed in the iceberg analysis. To do this, we used the RGB images with large zooms ($500 \times 500$ pixels) where we could adjust the contrast accordingly. The RGB images were composed with the intensities of the Pauli components: HH + VV for red, HH-VV for green and 2 HV for blue. An iceberg was selected if it represented an anomaly in backscattering with one (or all) of the following features: it was very close to the glaciers, it had a dark shadow in the looking direction (far range side), it had a bright rim in the direction of the sensor, it showed signs of breaking the sea ice surrounding it by being grounded and not able to move as the rest of the sea ice.

To keep a more conservative approach, other targets such as ships identified by a characteristic elongate shape were removed from the analysis. The same approach removed charter rocks or targets very large in size and with visual characteristics similar to islands.

Each target identified as an iceberg on the image was pinpointed by identifying the middle pixel in radar coordinates. Since we performed filtering, the middle pixel will be significant for an area of the iceberg, either $25.5 \times 21.5$ m or $56.1 \times 47.3$ m. Identification of the middle pixel was done because one of the running hypotheses of our analysis is that icebergs are a composition of different scattering mechanisms and if we would have averaged all the points together, we would not have been able to check for this. Only by keeping the area smaller were we able to perform multi-scale analysis.

It is important to point out that as we are missing in situ validation data, we are restricting our analysis to icebergs we can identify by visual analysis. The ones that cannot be visualised because of a small size or a wet surface (and therefore, very dark) are extremely challenging for the analysis, since they will require extensive in situ observations.

*2.2. Geographical Location and Meteorological Data*

The icebergs used in our analysis are all situated within the Greenland area. Greenland is influenced by various weather patterns, as well as the Gulf Stream and East Greenland Current,

which affects temperatures on the east and west sides. Blosseville Coast runs along the southeast of Greenland down towards Isortoq. Nuugaatsiaq sits on the west coast and Savissivik sits on the far northwest coast. Figure 2 shows the locations of the ALOS-2/PALSAR-2 images, while Table 3 includes the dates of each acquisition.

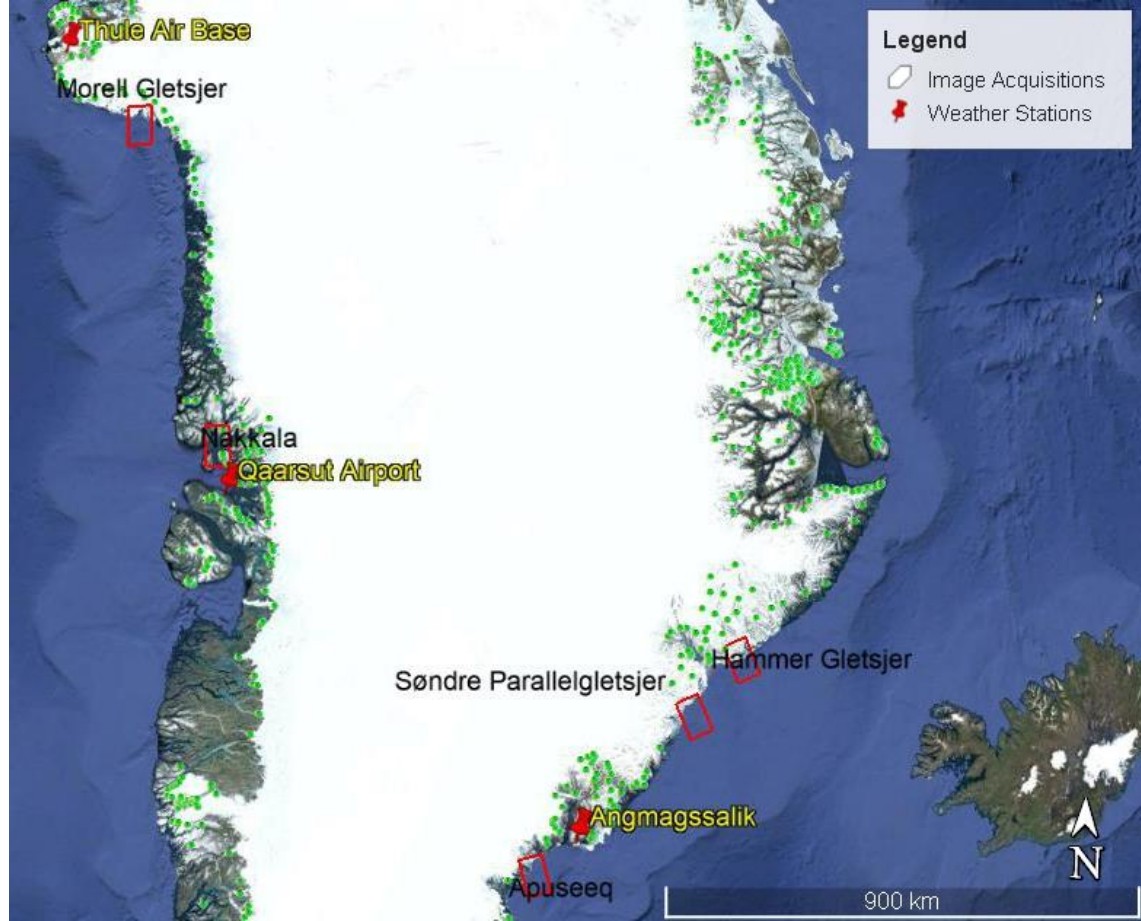

**Figure 2.** Google Earth image of data acquisition. Red pinpoints indicate the weather observation stations, with names shown in yellow labels. Red boxes indicate the image footprints. Green dots indicate GIS data for Greenland glacier locations. Glaciers names are in black. White compass point indicates north. 900 km indicates the scale of the image.

**Table 3.** Average Greenland meteorological conditions for images taken. Each location is a weather observation station. Note the image taken near Savissivik is acquired in December 2017, while the images taken near Isortoq and Nuugaatsiaq are taken in August 2015. Average data represent the month of image acquisition. Temperature is monthly averaged minimum.

| Location | Min Temperature (°C) | Average Rainfall (mm) | Average Wind Speed (km/h) | Date Taken |
|---|---|---|---|---|
| Angmagssalik | −3 | 5.44 | 6.9 | 03/08/2015 |
| Angmagssalik | −4 | 25.18 | 7.6 | 20/06/2015 |
| Angmagssalik | −3 | 5.44 | 6.9 | 15/08/2015 |
| Qaarsut Airport | 1 | 78.94 | 6.2 | 05/08/2015 |
| Thule Air Base | −18 | 4.83 | 11.7 | 06/12/2017 |

Because backscattering behaviour may be dependent on environmental factors such as the presence of surface liquid water, (besides size and shape of the icebergs) meteorological information such as temperature and precipitations were collated for the nearest available weather stations. These data

were also collected according to monthly data trends from 2015 and 2017. Three weather stations were investigated: Isortoq in east Greenland, Savissivik in north Greenland and Nuugaatsiaq in west Greenland. There is no weather data for the two Blosseville Coast locations, and we approximated these data using the closer Isortoq station.

### 2.3. Meteorological Conditions

The meteorological conditions in east, north and west Greenland, including temperature and precipitation levels, were also examined according to the locations of the three images. A record of the temperature and precipitation levels for Isortoq, Savissivik and Nuugaatsiaq corresponding to the dates of the image creation are shown in Table 3.

Due to the time of day the SAR images were taken (between 01:26 and 02:50 a.m.), minimum temperature was chosen for each location. There is a 19 °C difference between the locations. Average precipitation showed only rainfall for this period, with Nuugaatsiaq on average receiving 73.5 mm more than Isortoq. Average wind speeds did not very much, being around 6.5 km/h. The exception is Savissivik, which had a wind speed higher than that of the other two stations. Although we do not know the exact temperatures when the images were taken, these data are useful in showing that in some locations, such as Isortoq and Nuugaatsiaq, it is possible to observe the temperatures at which ice will melt.

Due to the effect of the Arctic polar days and nights, it is likely that in all of the images (with the exception of Savissivik) the iceberg surfaces may have been subject to solar radiation. This is because these images were acquired during summer months (Table 1). Radiation may have led to surface melting, in which the top of bergs may have some surface liquid water. The opposite is likely for Savissivik, as this image was taken in December and the effect of 24 h darkness together with a high wind speed and a very low temperature would have decreased the likelihood of surface liquid water present on the icebergs.

### 2.4. Glaciers that Calved Icebergs

Classification data of glaciers in Greenland are publicly available [34]. In this section, we have taken the approximate locations of each glacier and characteristics of icebergs that may have calved from them.

Table 4 outlines glacier names and geophysical parameters. The calving rates are based on an estimated retreat [35]. Glacier names are based off the dataset from [34] and the glacier tongue widths are calculated with distance tools in GIS software using images taken on 31 December 2016. Iceberg sizes are evaluated on SAR images and based on the classification detailed in [15] and approximates from each image.

**Table 4.** Glacier geophysical parameters. Note that the tongue widths, calving rates and iceberg sizes are estimates. Glacier names are taken from a database of known Greenlandic glaciers.

| Name of Glacier | Tongue Width (km) | Estimated Calving Rate (km/yr) | Location | Iceberg Size |
|---|---|---|---|---|
| Hammer | 16 | <5 | Blosseville Coast N | Small |
| Nakkala | 15 | <5 | Nuugaatsiaq | Medium |
| Apuseeq | 4 | <5 | Isortoq | Small |
| Søndre Parallelgletsjer | 12 | <5 | Blosseville Coast S | Large |
| Morell | 10 | <5 | Savissivik | Small |

In the following, we will investigate if the different size is resulting in different backscattering behaviour. Based on what we observe from SAR images, the tongue width does not seem to be easily related to the iceberg size. We also expect that the East Greenland Current could be having a significant effect on iceberg drift.

One important consideration is that icebergs small in size may be more prone to toppling over. This will show the saline ice surfaces that will be much less penetrable by the radar radiation. This effect is expected to be similar to having surface liquid water and it can result in making the iceberg less bright in images.

### 2.5. SAR Dataset

All the SAR data are quad-polarimetric, with a mean incidence angle of 34.9, an ascending mode and a right observation direction. A total of 1332 icebergs were identified in this paper. The way we identified the icebergs is by looking at small targets which stand out from the background. The main metric was to look at brightness, shape, presence of shadow (when visible) and the effect that the icebergs do not surround sea ice (if grounded). This means that the analysis will be inherently restricted to icebergs, which are visible in images. The smallest icebergs we identified were only a few pixels across, while the largest were a few tens of pixels. Clearly the size of the iceberg can be well bigger than the part that is visible in SAR images, and therefore, these values are only indicative of the iceberg part that is scattering substantially, but it is very likely that the whole iceberg is much bigger than this. This leads to a minimum visible size of around $10 \times 10$ m and a maximum of $215 \times 255$ m.

### 2.6. PolSAR

The Sinclair (S) scattering matrix [36], introduced in 1945 by G.W Sinclair, is used to characterise the polarimetric backscattering property of a target. The matrix can characterise a single target with a fixed polarimetric behaviour.

$$S = \begin{bmatrix} HH & HV \\ VH & VV \end{bmatrix} \tag{1}$$

H stands for linear-horizontal polarisation and V for linear-vertical polarisation. The repeated letter stands for Transmit–Receive. The transmission of a linear vertical wave, which is then received as a linear horizontal wave, gives HV. HH and VV are also referred to as co-channels, while HV and VH are cross-channels. We can also use a scattering vector $\underline{k}$ to characterise a polarised target:

$$\underline{k} = \frac{1}{2} Trace([S]\psi) = [k_1,\ k_2, k_3, k_4]^T \tag{2}$$

where *Trace* refers to the sum of all diagonal elements of a matrix and $\psi$ is a basis for a $2 \times 2$ Hermitian matrix. $k_1$, $k_2$ and $k_3$ are complex numbers. In the case of a monostatic sensor or a reciprocal medium, HV = VH, except for noise and $\underline{k}$, becomes a three-dimensional complex. We define scattering mechanism or projections vector as a normalised $\underline{k}$ vector [37].

$$\underline{\omega} = \frac{\underline{k}}{|\underline{k}|} \tag{3}$$

Acquiring the full scattering matrix provides quad-polarimetric (quad-pol) data. The data are referred to as dual-polarimetric if there are only two polarisation channels which are coherent. In SAR, targets tend to be distributed or composed of numerous objects. This is also the cause of speckle (multiplicative noise). A full review of speckle is available in [38,39]. In this case, it is very likely that every pixel contains a different polarimetric behaviour, which a single scattering matrix cannot characterise. A distributed target composed by several single targets is referred to as a partial target [37]. We therefore extract the second order statistics by evaluating the $3 \times 3$ covariance matrix, which can be estimated as:

$$|C| = \langle \underline{k} \cdot \underline{k}^{*T} \rangle = \begin{bmatrix} \langle |k_1|^2 \rangle & \langle k_1 k_2^* \rangle & \langle k_1 k_3^* \rangle \\ \langle k_2 k_1^* \rangle & \langle |k_2|^2 \rangle & \langle k_2 k_3^* \rangle \\ \langle k_3 k_1^* \rangle & \langle k_3 k_2^* \rangle & \langle |k_3|^2 \rangle \end{bmatrix} \tag{4}$$

where <> is an averaging operator, * is a complex conjugate and T is the matrix transpose. Each element of the covariance matrix is made up of combinations of the components of $\underline{k}$.

Target decomposition theorems are used to interpret polarimetric information in the scattering matrix, or the covariance matrix. They were initially introduced in the application of light scattering by Chandrasekhar [40] but later became used for scattered waves [41].

SAR images present speckle and before a target decomposition theorem can be applied, speckle noise must be reduced by using filters. In this work, we use a standard boxcar filter [39] to average all the pixels inside a moving window because this provides results that are easier to interpret in our multi-scale analysis. We also use incoherent decompositions since the behaviour expected by icebergs and sea ice is as distributed partial targets, which present some form of statistical fluctuation. The averaging will help avoid misclassifications due to this random fluctuation of the polarimetric characteristics from speckle (regardless if it is fully formed or not).

### 2.6.1. Cloude–Pottier Decomposition

The Cloude–Pottier decomposition has been widely used to classify partial targets. The decomposition considers the diagonalisation of the coherency matrix (T), which is a covariance matrix when the Pauli basis is used. For a description of the Pauli basis, please see page 10. Since (T) is Hermitian, it has real eigenvalues and orthogonal eigenvectors [42]. Entropy is one of the parameters using the eigenvalues and it determines if depolarisation is happening. Entropy is represented by this equation:

$$H = -\sum_{i=1}^{3} P_i log_3(P_i) \tag{5}$$

We also determined $P_i$ such that:

$$P_i = \frac{\lambda_i}{\sum_{j+1}^{n} \lambda_j} \tag{6}$$

where $\lambda_i$ are the eigenvalues.

A characteristic angle ($\alpha$) for the scattered medium can be derived by using the eigenvector matrix in the following form:

$$|T| = [U_3] \begin{bmatrix} \lambda_1 & 0 & 0 \\ 0 & \lambda_2 & 0 \\ 0 & 0 & \lambda_3 \end{bmatrix} |U_3|^{*T} \ [U_3] = \begin{bmatrix} \cos\alpha_1 & \cos\alpha_2 & \cos\alpha_3 \\ \sin\alpha_1\cos\beta_1 e^{i\delta_1} & \sin\alpha_2\cos\beta_2 e^{i\delta_2} & \sin\alpha_3\cos\beta_3 e^{i\delta_3} \\ \sin\alpha_1\sin\beta_1 e^{i\gamma_n} & \sin\alpha_2\sin\beta_2 e^{i\gamma_2} & \sin\alpha_3\sin\beta_3 e^{i\gamma_3} \end{bmatrix} \tag{7}$$

Next, the target is modelled as three separate (S) matrices occurring with probabilities of $P_i$, so that ($P_1 + P_2 + P_3 = 1$). The mean characteristic alpha angle is produced by the three $\alpha_i$ parameters in random sequences. Alpha angle is represented by this equation:

$$\alpha = p_1\alpha_1 + p_2\alpha_2 + p_3\alpha_3 \tag{8}$$

The parameter alpha is common in radar polarimetry and is used to determine the type of scattering present while it sweeps from 0° to 90°: odd-bounce, dipoles and even bounce. The parameter beta (β) is used to determine the orientation angle of a target about the line of sight.

Boxcar filters produce optimum results when a homogenous surface is involved. For our multi-scale analysis, we used it with a $5 \times 5$ and an $11 \times 11$ averaging window.

For each image, we produced a Pauli RGB image based on the Pauli basis. The Pauli basis allows for the separation of odd bounce, even bounce and 45° tilted bounce components. It is applied in the case of a monostatic sensor as a 3D lexicographic vector:

$$L = \begin{bmatrix} S_{HH} & \sqrt{2S_{HV}} & S_{VV} \end{bmatrix}^T \tag{9}$$

The Pauli representation can then be presented as a 3D Pauli vector:

$$\underline{k} = \frac{1}{\sqrt{2}}[S_{HH} + S_{VV} \ \ S_{HH} + S_{VV} \ \ 2S_{HV}]^T \tag{10}$$

Total backscattered power of the Pauli vector or lexicographic feature vector (span) is then calculated. We apply span to both averaging windows after the majority of speckle is boxcar filtered.

$$Span = \left\langle |S_{HH}|^2 \right\rangle + 2\left\langle |S_{HV}|^2 \right\rangle + \left\langle |S_{VV}|^2 \right\rangle \tag{11}$$

### 2.6.2. Yamaguchi Decomposition

Several studies have applied and adapted the three-component scattering model devised by Freeman and Durden [43]. In this paper, we used the Yamaguchi decomposition that is also derived and modified from the Freeman–Durden model. The scattering model used here consists of four scattering mechanisms (double bounce, surface, volume and helix) [44]. The decomposition uses the covariance matrix. The simplicity of this model is what makes it suitable for quick and easy image processing. The model has been successfully applied to C-band [45], L-band [44] and P-band SAR data [46]. The main difference with Freeman is that Yamaguchi noted that this scattering model is limited, particularly in urban areas due to the reflection symmetry assumption which does not consider asymmetric scattering behaviours. For this reason, a fourth mechanism (helix) is included. Additionally, the model applies a removal of the orientation angle before extracting the value for volume (this avoids overestimation due to the fact that HV is representative of 45° oriented dihedrals).

## 3. Results

### 3.1. Preliminary Image Analysis

Figure 3 presents the Pauli RGB image for Blosseville Coast N. The RGB Pauli component represents the elements of the Pauli vectors in linear scale. Since the scale does not allow us to see details, Figure 4 presents a visual close-up identification of an area of the image with an abundance of icebergs. The figure also includes the corresponding alpha and entropy for both averaging window sizes. Looking at the Pauli RGB, it is easy to identify icebergs in the channel area. The image was acquired in August. The sea area seems to be a mix of open ocean, thin sea ice (bluish areas), small floes and icebergs. Please note how the bright points (generally with shadow features) appear to generally have a low alpha angle and a rather low entropy, which would suggest surface scattering mixed to volume scattering. Comparing $11 \times 11$ and $5 \times 5$, we can see the feature seems to be mostly consistent in these images, but the following plots will show that some differences exist.

Figure 5 presents the Pauli RGB images with a corresponding alpha and entropy image for the $5 \times 5$ window for Nuugaatsiaq and the visual analysis of icebergs. This acquisition was also done in August and the sea appears mostly as open ocean. Again, icebergs are relatively easy to identify in the alpha image as low values with surface scattering and a medium entropy.

Figure 6 presents the Pauli RGB images with a corresponding alpha and entropy image for the $5 \times 5$ window for Isortoq as well as the visual analysis of the icebergs. This is also an August image and the sea region is a mix of open ocean (bluish area) and small flows (grey areas).

Finally, Figure 7 presents the Pauli RGB for Blosseville Coast S and Savissivik and the visual analysis of the icebergs in each. The first was acquired in June and the other in December. The area here is clearly covered by sea ice.

Note that in some images, there are red spots and stripes in the sea regions. These are due to bright azimuth ambiguities from the mountains near the coastline. In all the Pauli images, icebergs appear as bright points, mostly white (due to saturation in visualisation). In some images, sea ice can be rather bright and appear in grey. The lack of colour is likely to be due to unpolarised scattering

where the three components of the Pauli are similarly weighted. This can happen in ice with large deformation features or (more likely in our case) with very small floes with high rims (as pancakes).

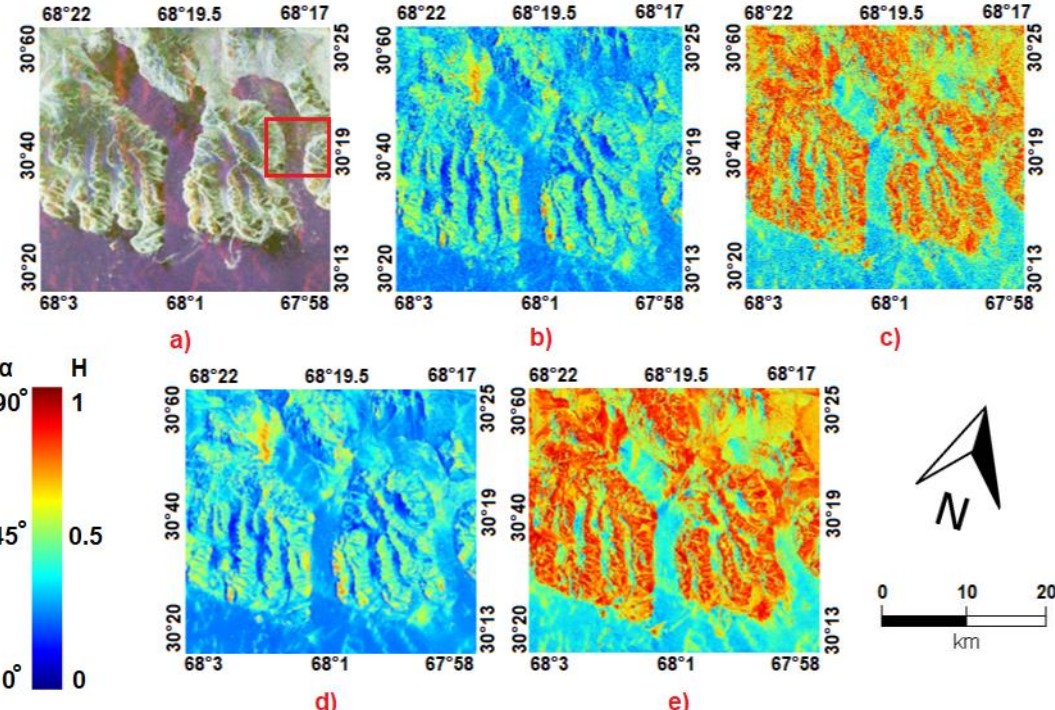

**Figure 3.** Output images for Blosseville Coast N, 15/08/2015 01:26, (**a**) Pauli RGB image, (**b**) alpha image in a 5 × 5 window, (**c**) entropy image in a 5 × 5 window, (**d**) alpha image in an 11 × 11 window, (**e**) entropy image in an 11 × 11 window. Red box indicates extent of Figure 4. Numbers on the edges of images indicate image DMS coordinates. The sea ice situation is a mix of ice floes and open ocean.

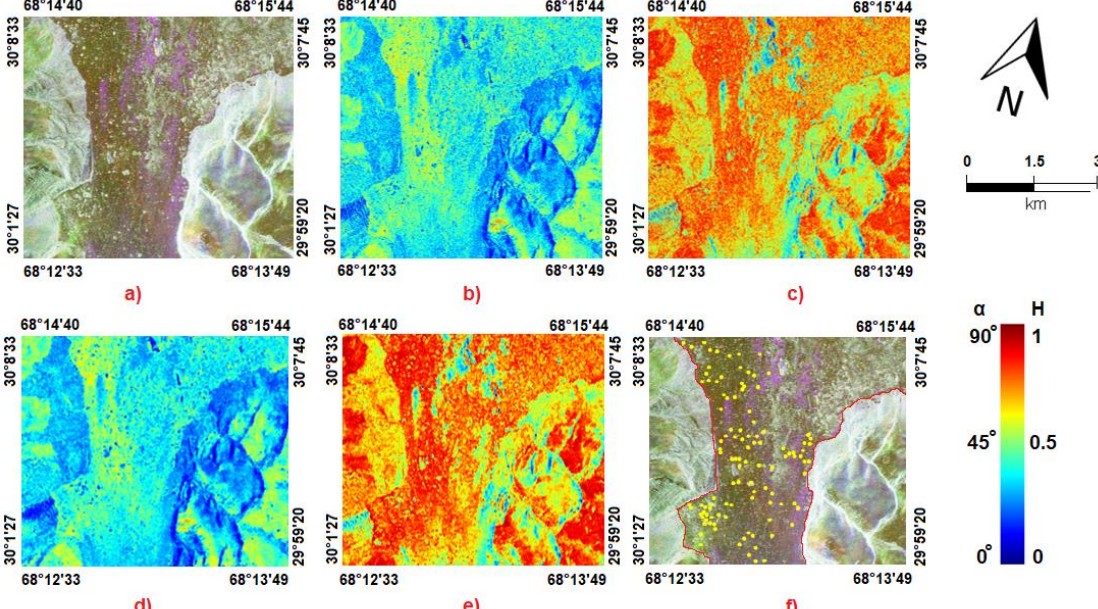

**Figure 4.** Output close-up images for Blosseville Coast N, 15/08/2015 01:26, (**a**) Pauli RGB image, (**b**) alpha image in a 5 × 5 window, (**c**) entropy image in a 5 × 5 window, (**d**) alpha image in an 11 × 11 window, (**e**) entropy image in an 11 × 11 window, (**f**) visual analysis of icebergs (in yellow) and coastline (in red). Numbers on the edges of images indicate image DMS coordinates. The sea ice situation is a mix of ice floes and open ocean.

Although most of the icebergs appear as blue spots (showing surface scattering), few especially in Blosseville Coast N appear as red spots showing double bounce. This following quantitative analysis will show that icebergs are more commonly showing either surface or volume scattering and only rarely, double bounce.

In Blosseville Coast S, we were able to clearly observe icebergs only in the open water region. There are few large icebergs in Savissivik with smaller icebergs in the surroundings.

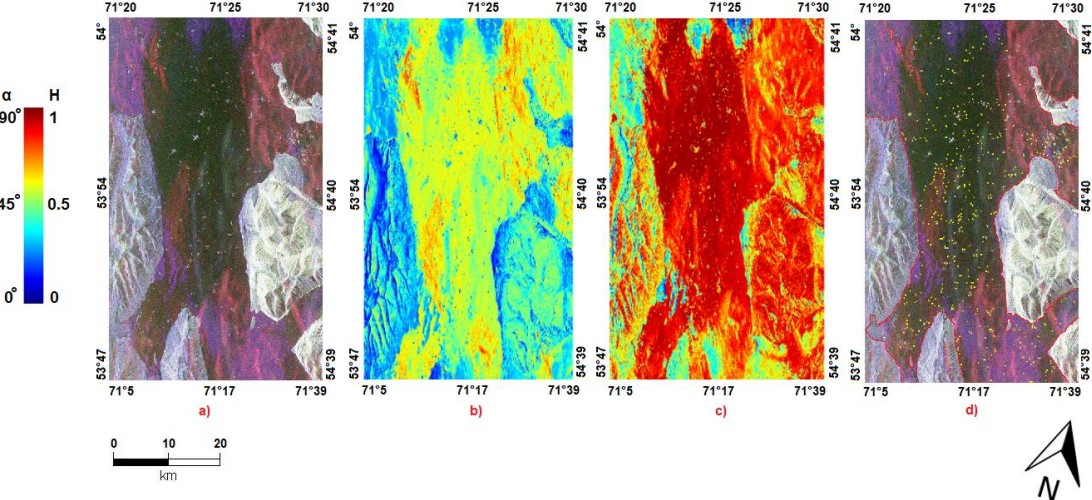

**Figure 5.** Output images for Nuugaatsiaq, 05/08/2015 02:48, (**a**) Pauli RGB, (**b**) alpha image in a 5 × 5 window, (**c**) entropy image in a 5 × 5 window, (**d**) Pauli RGB visual analysis of icebergs. Yellow dots show icebergs. Red outline indicates coastline. Numbers on the edges of images indicate image DMS coordinates. The sea ice situation is mostly open ocean.

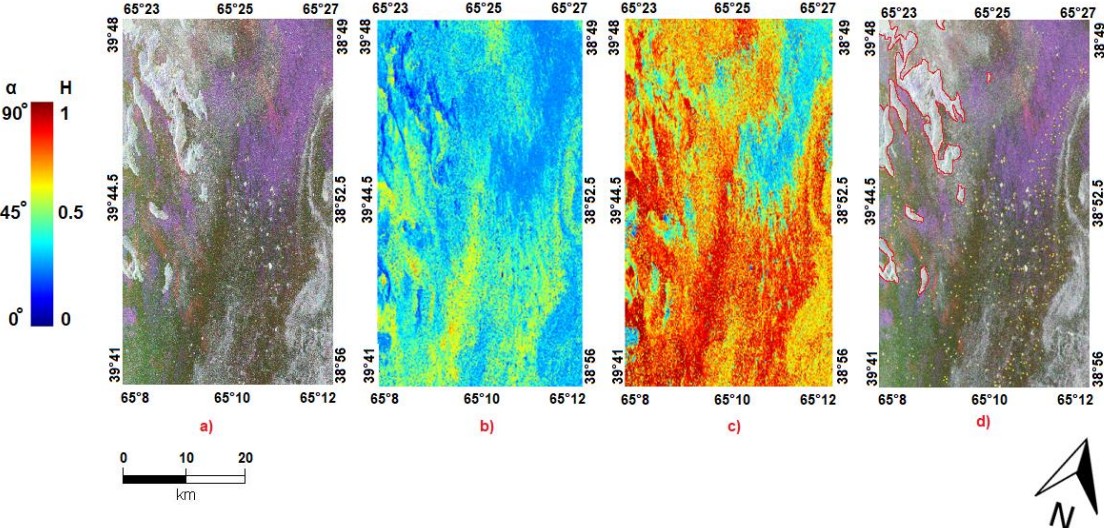

**Figure 6.** Output images for Isortoq, 03/08/2015 02:07, (**a**) Pauli RGB, (**b**) alpha image in a 5 × 5 window, (**c**) entropy image in a 5 × 5 window. (**d**) Pauli RGB visual analysis of icebergs. Yellow dots show icebergs. Red outline indicates coastline. Numbers on the edges of images indicate image DMS coordinates. The sea ice situation is a mix of open ocean (bluish area in the RGB) and small flows (grey areas).

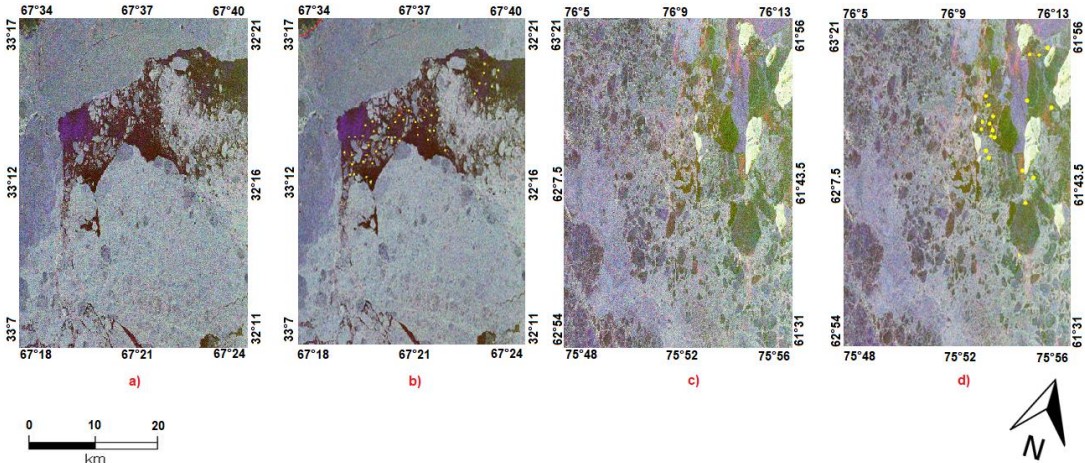

**Figure 7.** Output images for (**a**) Blosseville Coast S Pauli RGB image 20/06/2015 01:26, (**b**) Blosseville Coast S visual analysis, (**c**) Savissivik RGB image 06/12/2017 02:52, (**d**) Savissivik visual analysis. Yellow dots show icebergs. Red outline indicates coastline. Numbers on the edges of images indicate image DMS coordinates. The sea ice situation is mostly pack ice with several leads.

### 3.2. Polarimetric Behaviour

The following analysis uses plots to extract information about the polarimetric behaviour of icebergs and to compare this across the different locations. The colours in the images represent the different locations of icebergs (i.e., the five images).

In order to assess how spatially packed the scatterers are in icebergs, we performed a multi-scale analysis where two moving windows were used, a $5 \times 5$ and an $11 \times 11$. One of the objectives is to understand if icebergs properly fit the model of partial targets (where the single target components are uniformly distributed) or if they are a composition of single targets located in proximity to each other.

### 3.2.1. Cloude–Pottier

In this section, we present the Cloude–Pottier boxplots and scatter plots for analysis. The first parameter we want to analyse is the entropy. An important factor when evaluating the entropy is to check its relation to the overall brightness, since this is also an indicator of the presence of dominant scatterers or the closeness of the backscattering to the noise floor (which will increase the value of entropy).

The boxplots for the entropy are presented in Figure 8, where the two subplots show the change in values with increasing window size. It is very interesting to notice that icebergs present a wide range of entropy values. The $11 \times 11$ window, which is the most commonly used for this, shows that we have low entropy (single targets) for the icebergs in Savissivik, Blosseville Coast N and S, while other locations present mid to high levels of entropy. Isortoq presents relatively high entropy and lower backscattering, suggesting that there are no very dominant strong scatterers in those icebergs.

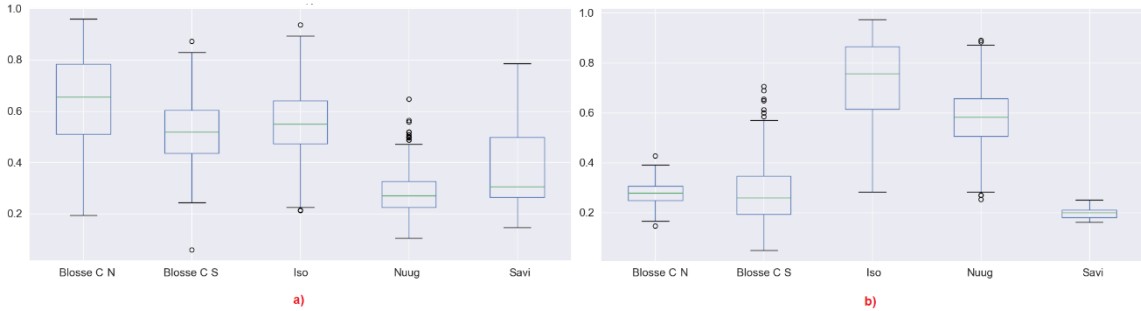

**Figure 8.** (**a**) Iceberg entropy boxplot $5 \times 5$ window, (**b**) $11 \times 11$ window. Large entropy changes are most significant in Blosseville Coast N and Savissivik. Dots indicate outliers.

Please note how the values change when modifying the window size. This will be discussed in the following section.

The value of the entropy can be influenced by the span, and therefore, we first present the boxplot of the span in Figure 9, and then, in Figure 10, we show the span against entropy in a $5 \times 5$ averaging window and an $11 \times 11$ window. We can see that lower values of span (below $-20$ dB) have generally higher values of entropy (above 0.5). This is in line with the fact that darker icebergs are closer to the noise floor, which adds depolarisation.

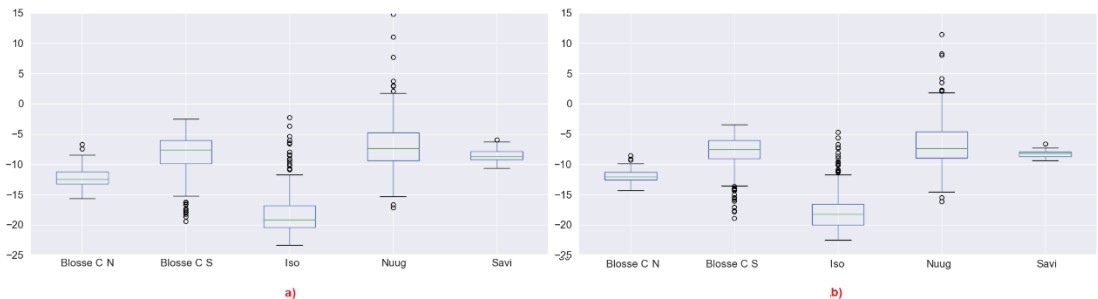

**Figure 9.** (**a**) Iceberg span boxplot $5 \times 5$ window, (**b**) $11 \times 11$ window. Large entropy changes are most significant in Blosseville Coast N and Savissivik. Dots indicate outliers.

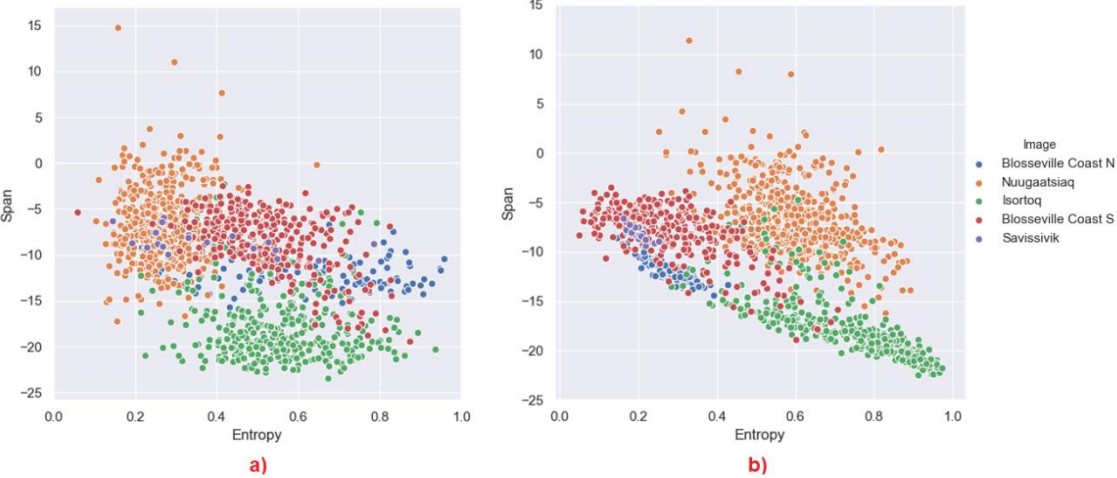

**Figure 10.** (**a**) Iceberg entropy, span plot $5 \times 5$ window, (**b**) $11 \times 11$ window. Note the negative values for span. Entropy values are between 0 and 1. Colour legend indicates each image and each dot is an iceberg.

We can now proceed analysing alpha. The boxplots for the averaged alpha are presented in Figure 11, where again, the two subplots show the change in values with modifying window size. In order to have a more meaningful interpretation of the alpha values, Figure 12 shows the averaged alpha angle against entropy in a $5 \times 5$ window and an $11 \times 11$ window. The average alpha is quite low when the entropy is low. However, there are few points where the alpha reaches high values. Clearly, when the entropy increases, the alpha is forced toward $60°$.

Figure 13 shows that there is no pattern between alpha and span, beside the fact that Isortoq has low values of span, which leads to a higher entropy, and therefore, a higher alpha.

Another parameter of interest is beta. The boxplot of beta is presented in Figure 14, while the plots in Figure 15 show the average beta against alpha angle for $5 \times 5$ and $11 \times 11$. Interestingly, we can see that ordinarily there is no preferred beta value. However, for some locations and window sizes, the beta angle seems to be backed around zero, indicating horizontally oriented scatterers. Since the

scattering is mostly surface, the impact of beta angle (present in the second and third element of the Pauli vector) is possibly noisy and more generally hard to interpret.

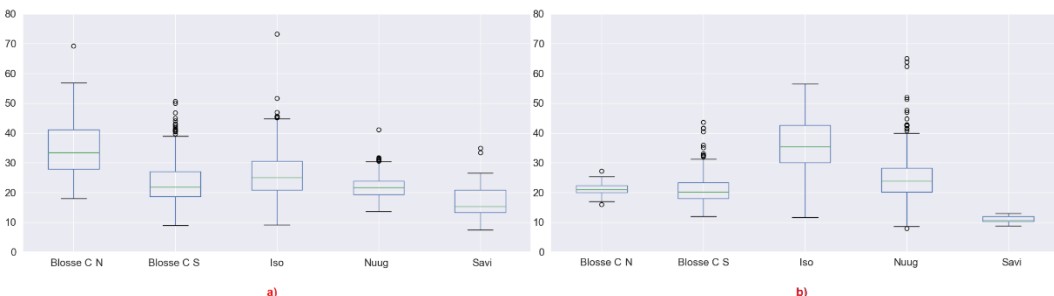

**Figure 11.** (**a**) Iceberg alpha boxplot 5 × 5 window, (**b**) 11 × 11 window. Significant changes in alpha are evident in Blosseville Coast N and Savissivik.

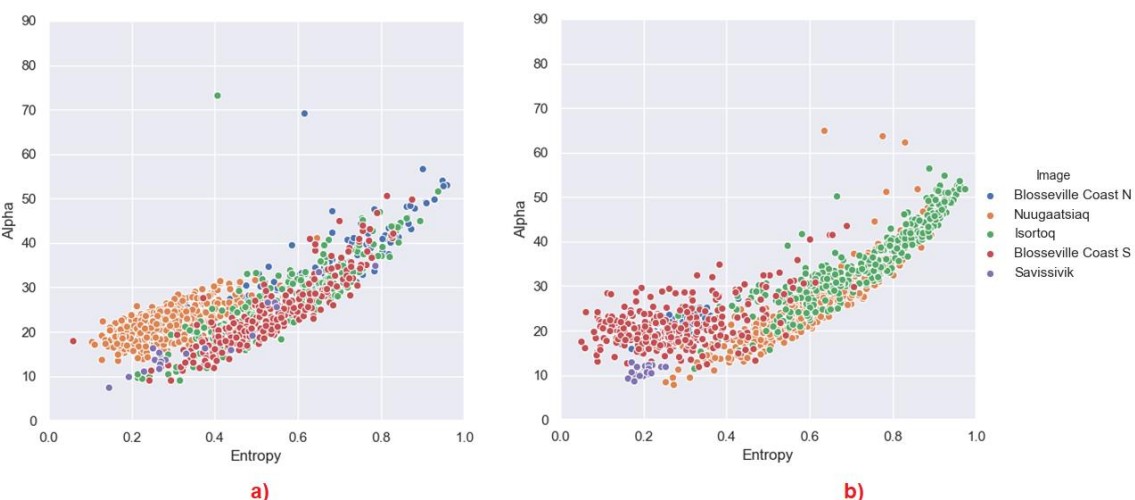

**Figure 12.** (**a**) Iceberg alpha, entropy plot 5 × 5 window, (**b**) 11 × 11 window. Entropy is between 0 and 1. Alpha is between 0 and 90. Colour legend indicates image. Dots indicate icebergs.

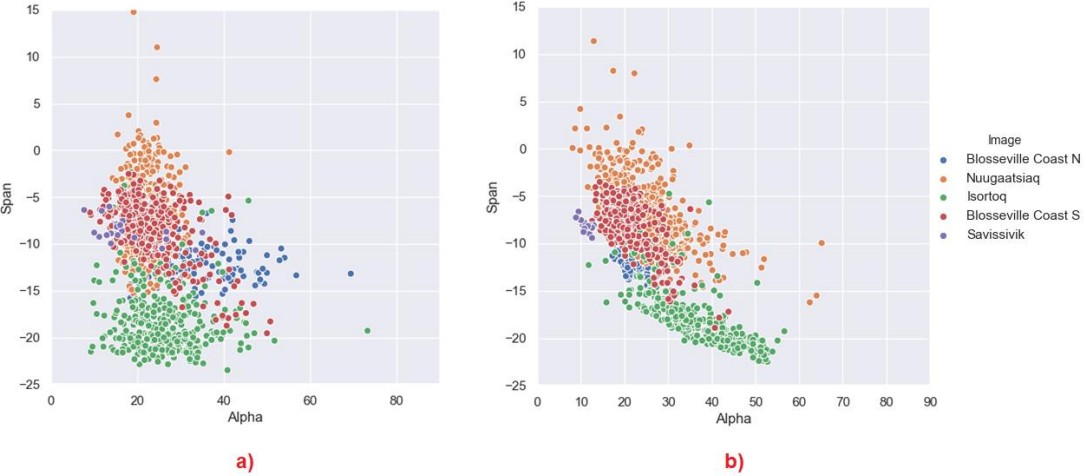

**Figure 13.** (**a**) Iceberg alpha, span plot 5 × 5 window, (**b**) 11 × 11 window. Colour legend indicates image. Dots indicate icebergs.

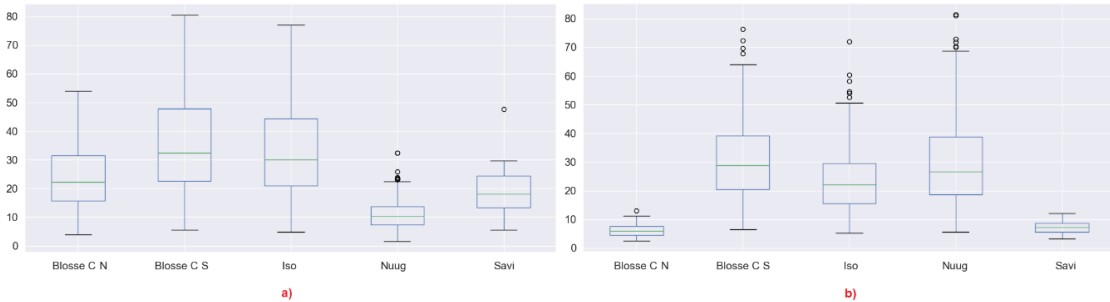

**Figure 14.** (**a**) Iceberg beta boxplot 5 × 5 window, (**b**) 11 × 11 window. Significant changes in beta are evident in Blosseville Coast N, Nuugaatsiaq and Savissivik.

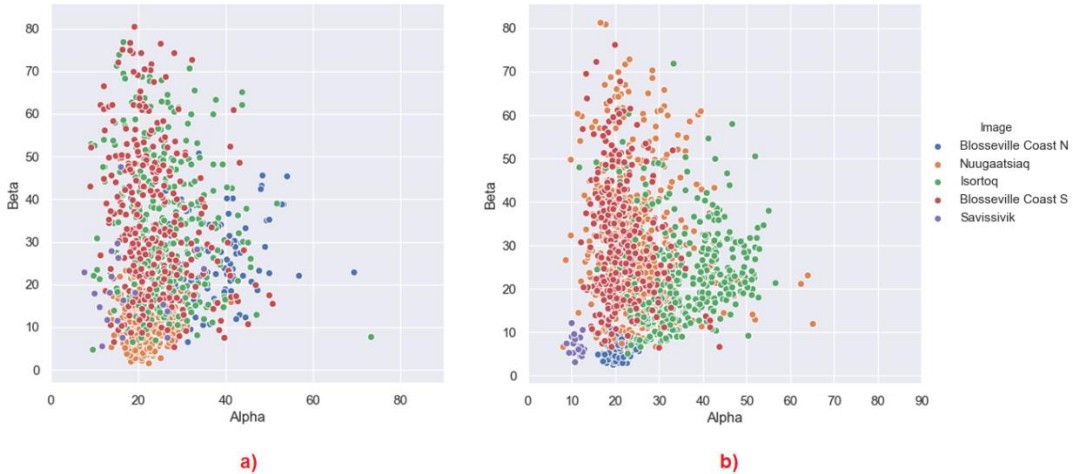

**Figure 15.** (**a**) Iceberg alpha vs. beta plot 5 × 5 window, (**b**) 11 × 11 window. Colour legend indicates image. Dots indicate icebergs.

Figures 16 and 17 shows the entropy against anisotropy. This is important to evaluate if there is only one or more scattering mechanisms besides the dominant one.

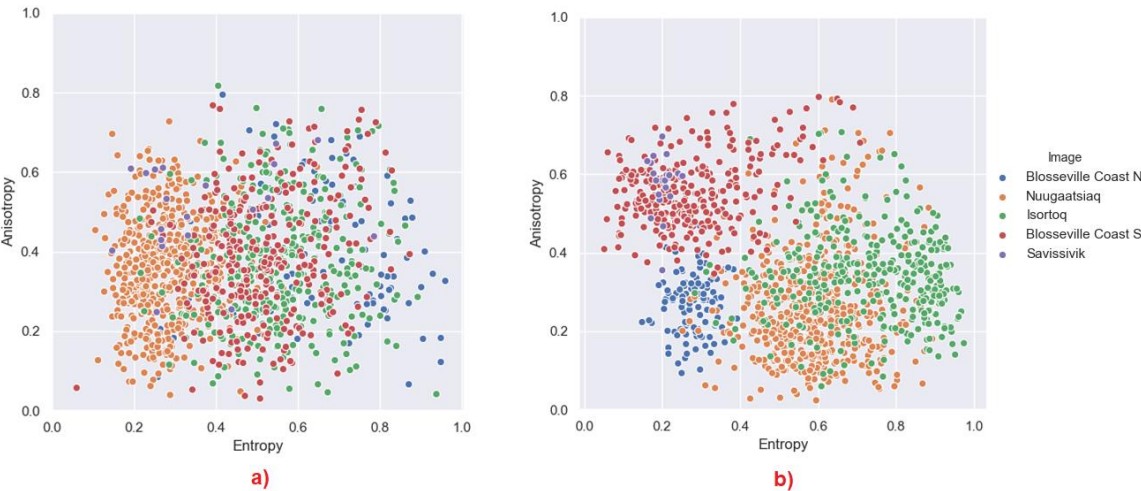

**Figure 16.** (**a**) Iceberg anisotropy boxplot 5 × 5 window (**b**) 11 × 11 window. Slight changes in beta are evident in Blosseville Coast N, and Savissivik.

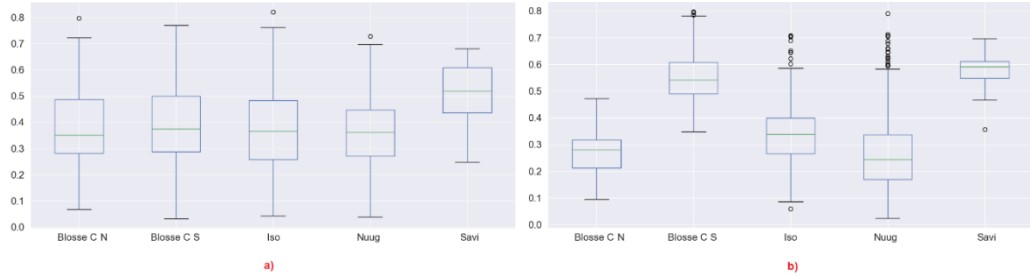

**Figure 17.** (**a**) Iceberg entropy, anisotropy plot 5 × 5 window, (**b**) 11 × 11 window. Colour legend indicates image. Dots indicate icebergs.

No evident trend appears here. However, the change with window size will be discussed later.

### 3.2.2. Yamaguchi

We also tested a four component scattering model devised by Yamaguchi [44]. The four components are double bounce, surface, volume and helix scattering. In Figures 22–27, we show dB plots between the different four components. A red line bisects through each graph, helping to distinguish if one of the two components is dominant. Please note, the algorithm we used avoids instability by clipping low values to the lowest value in the image. This is the reason why we can see repeated lowest values in the plots.

Here, we present the Yamaguchi boxplots for analysis (Figures 18–21). These are interesting to monitor. However, what is even more interesting is how the different components balance with each other. For this task, we produced scatter plots.

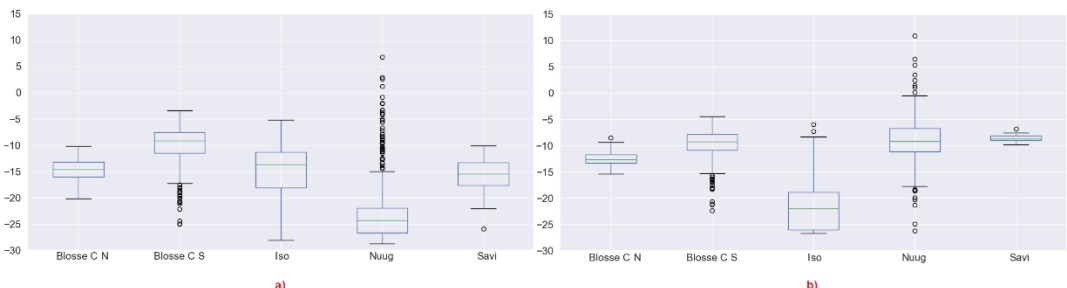

**Figure 18.** (**a**) Iceberg surface scattering boxplot 5 × 5 window, (**b**) 11 × 11 window. There are significant changes in surface scattering in Savissivik.

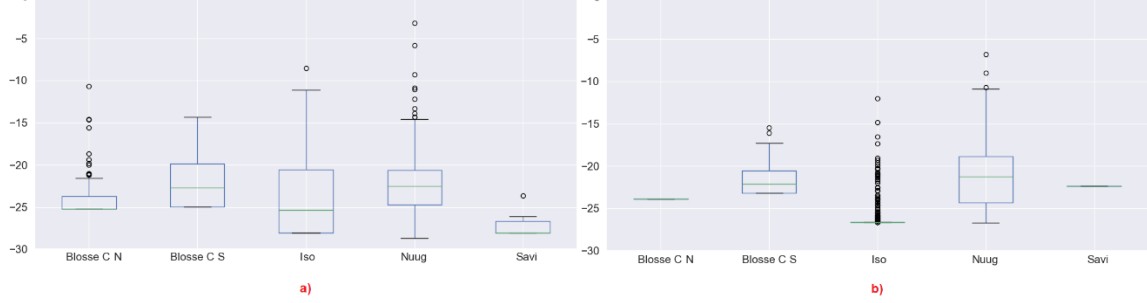

**Figure 19.** (**a**) Iceberg volume scattering boxplot 5 × 5 window, (**b**) 11 × 11 window. There are significant changes in volume scattering in Isortoq and Savissivik.

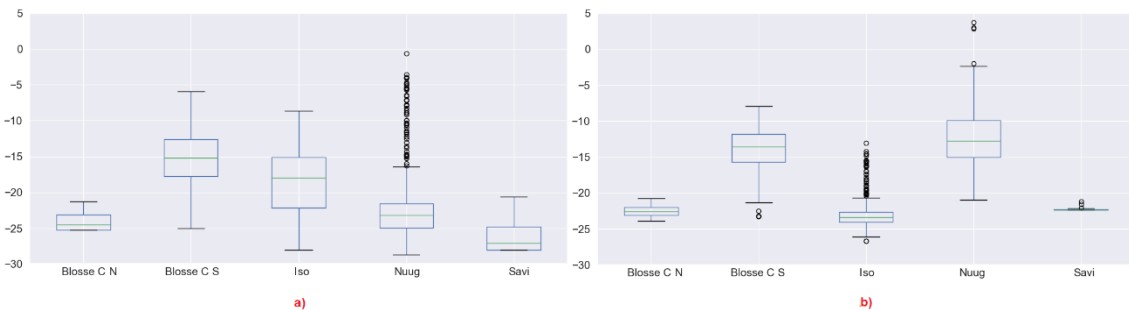

**Figure 20.** (**a**) Iceberg double bounce scattering boxplot 5 × 5 window, (**b**) 11 × 11 window. There are significant changes in Blosseville Coast N, Isortoq and Savissivik.

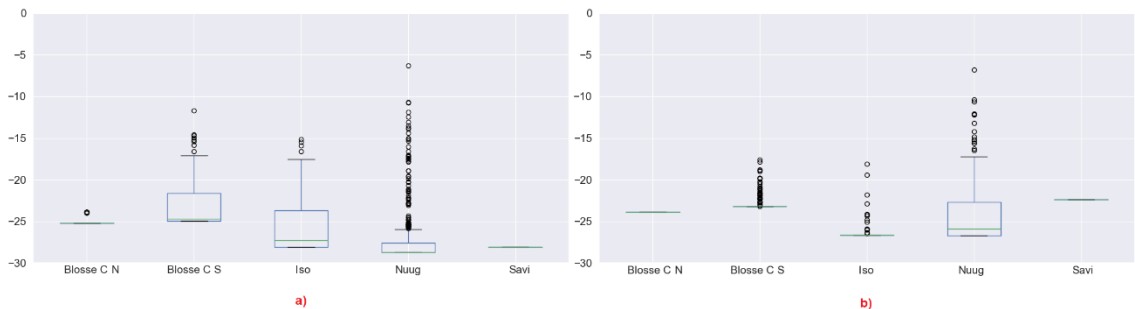

**Figure 21.** (**a**) Iceberg helix scattering boxplot 5 × 5 window, (**b**) 11 × 11 window. There are significant changes in helix scattering in Blosseville Coast S, Isortoq and Nuugaatsiaq.

From Figure 22, we can see that for icebergs in almost all scenarios, when the backscattering is high enough, surface scattering is larger than the volume scattering. This is also corroborated by the previous plots. However, when the signal is low, the difference is less evident. This may indicate that the volume scattering, although present, may not be the dominant mechanism for these icebergs at L-band.

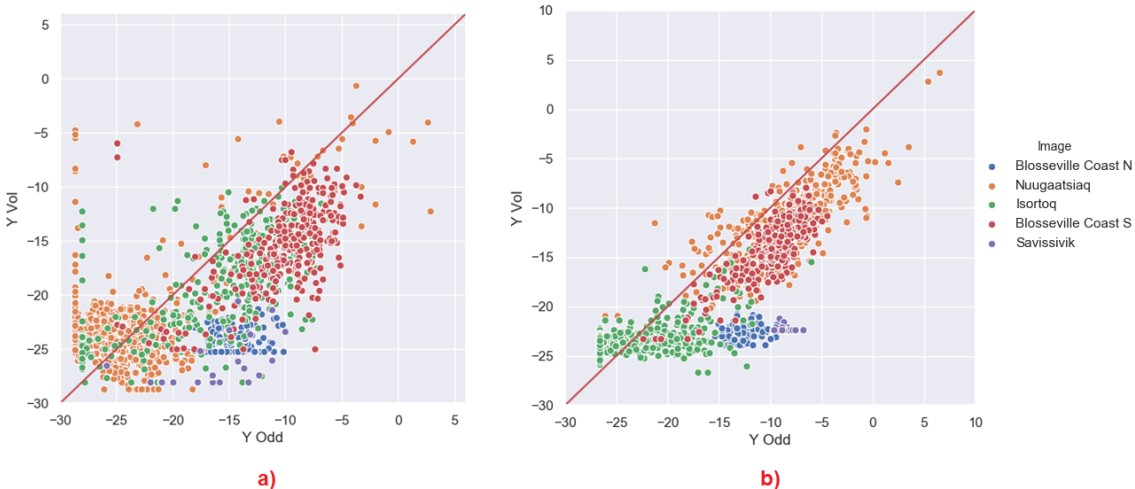

**Figure 22.** (**a**) Iceberg volume scattering, surface scattering plot in a 5 × 5 window, (**b**) 11 × 11 window. The majority of icebergs show surface scattering. Colour legend indicates image. Dots indicate icebergs. All values are in dB.

Figure 23 shows the comparison between volume and double bounce.

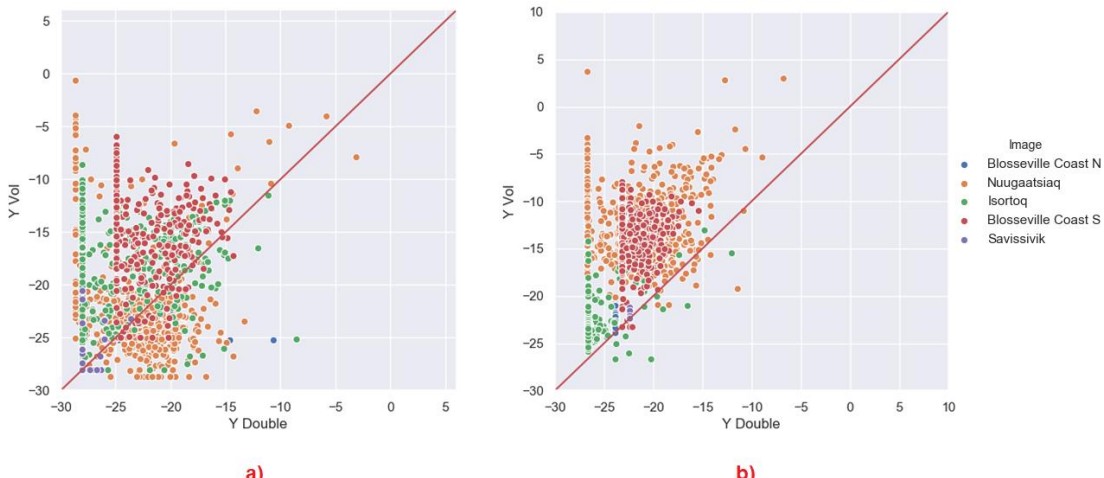

**Figure 23.** (**a**) Iceberg double bounce scattering, volume scattering plot in a 5 × 5 window, (**b**) 11 × 11 window. The majority of icebergs show volume scattering. Colour legend indicates image. Dots indicate icebergs. All values are in dB.

If we look at Figure 24, we compare the double bounce with surface. It appears as surface is again dominant in most of the icebergs with 5 × 5, except a few exceptions where the double bounce is stronger. Double bounce seems to be dominant only in a limited number of icebergs.

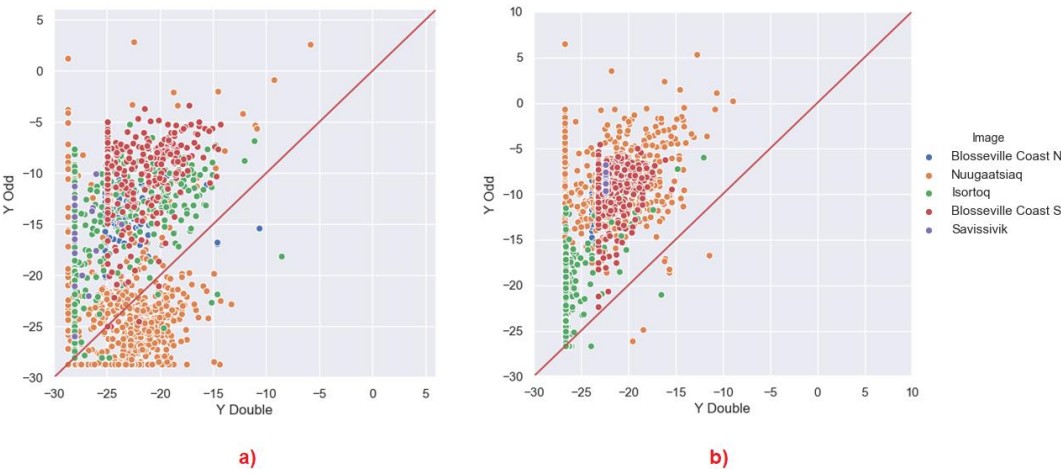

**Figure 24.** (**a**) Iceberg double bounce scattering, surface scattering plot in a 5 × 5 window, (**b**) 11 × 11 window. The majority of icebergs show volume scattering. Colour legend indicates image. Dots indicate icebergs. All values are in dB.

The following three plots in Figures 25–27 compare the helix scattering. We can see that helix scattering is generally not dominant compared to any other scattering mechanisms in all the locations. This has some exception for a few icebergs, but normally, the helix is not expected to be strong, showing a scattering behaviour which is relatively reflection symmetric.

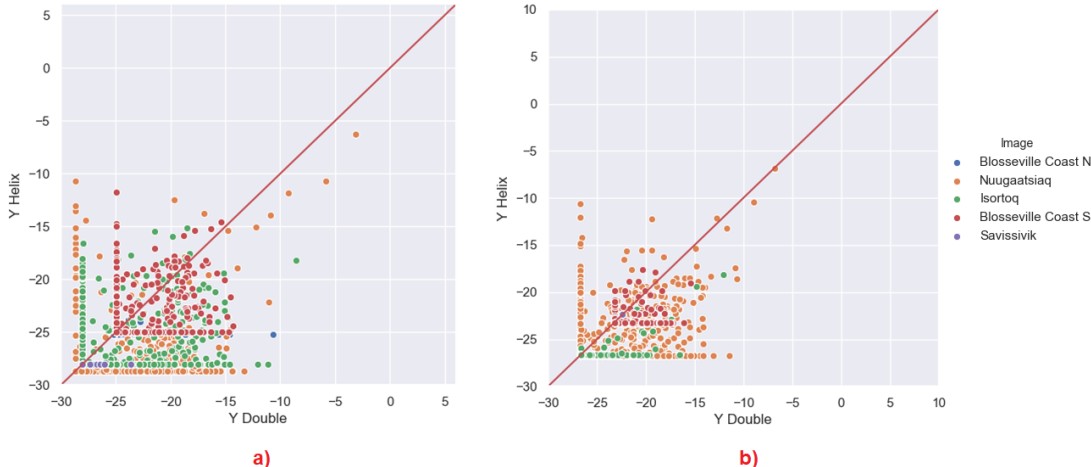

**Figure 25.** (**a**) Iceberg double bounce scattering, helix scattering plot in a $5 \times 5$ window, (**b**) $11 \times 11$ window. The majority of icebergs tend to show significant double bounce. Colour legend indicates image. Dots indicate icebergs. All values are in dB.

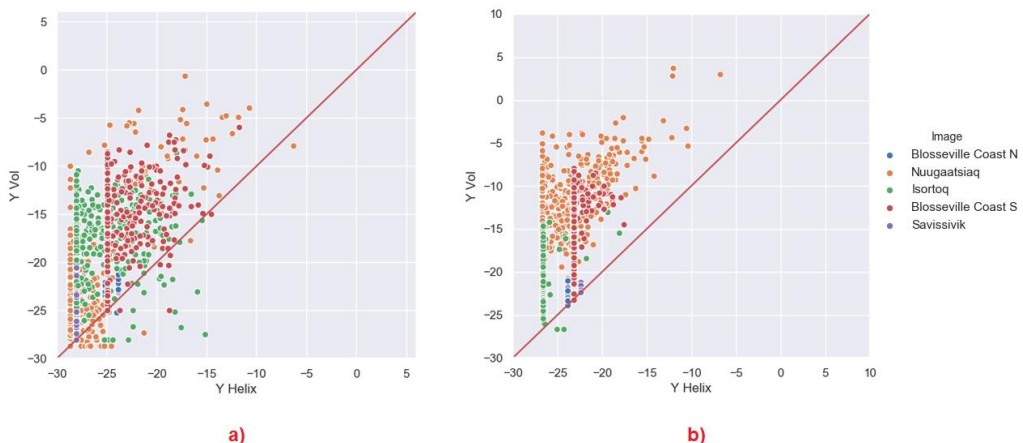

**Figure 26.** (**a**) Iceberg helix scattering, volume scattering plot in a $5 \times 5$ window, (**b**) $11 \times 11$ window. The majority of icebergs show more volume scattering. Colour legend indicates image. Dots indicate icebergs. All values are in dB.

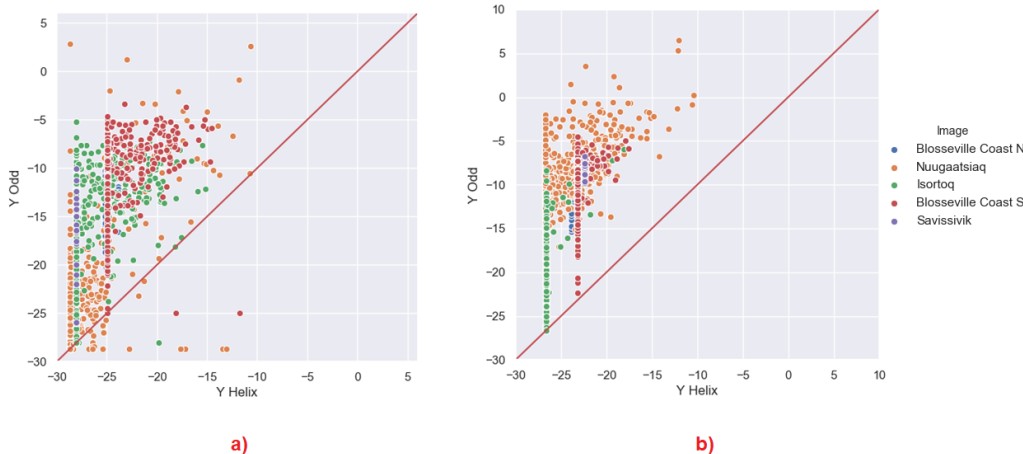

**Figure 27.** (**a**) Iceberg helix scattering, surface scattering plot in a $5 \times 5$ window, (**b**) $11 \times 11$ window. The huge majority of icebergs show surface scattering. Colour legend indicates image. Dots indicate icebergs. All values are in dB.

## 4. Discussion

The analysis of 1332 icebergs in Greenland showed that there are several commonalities in the scattering behaviours, but also differences. In the following, we identify the main highlights of this analysis.

### 4.1. Depolarisation

Entropy values on any given stationary target captured by SAR give a good indication of the amount of scattering mechanisms present within the target. The icebergs analysed seem to have a large variety of entropy values. We also performed a multi-scale analysis to check if icebergs can be approximated as (a) partial targets with a uniform distribution of scattering mechanisms, (b) single targets with a single scattering mechanism or (c) a mixture of single targets in close proximity. Looking at the differences in entropy, when the window was changed from $5 \times 5$ to $11 \times 11$, we can conclude that icebergs can follow in all the three categories. Even more interesting, the geographical location seems to suggest in which category they are. If the target would be a fully uniform partial target, then the entropy should not change, except that when we go with very small windows (e.g., $3 \times 3$), we may have a bias toward smaller values.

Interestingly, we can see two trends when reducing the window size to $5 \times 5$:

(a)  We may expect that by reducing the window, the entropy may tend to increase because in a smaller window, we expect that it is less likely to have lots of dominant scatterers. This is the case for instance for Nuugaatsiaq and partially for Isortoq. This is an indicator that dominant scatterers in these icebergs are not packed uniformly and very close to each other. When we use a smaller window, we include less dominant scatterers and increase the entropy. Considering the window sizes, we may expect strong scatterers being located no closer than a few tens of metres. This may represent some topographic features of the iceberg. From a more theoretical point of view, it indicates that scattering from those icebergs is not well approximated by a partial target with fully developed speckle. A uniform distribution of scatterers is slightly more realistic for Isortoq, where the reduction is not very large, although the values are already quite high to start with due to the low backscattering and the effect of noise.

(b)  Blosseville Coast N and S and Savissivik are an interesting case, since several icebergs reduce their entropy when increasing the window. Inspecting the images, we revealed that these are smaller icebergs and when increasing the window, we included the edge pixels, which are generally brighter. We therefore included in the window other dominant scatterers that increase the entropy. Please note, we used the middle pixels of the icebergs because we are interested in the scattering behaviour of the ice body, in order to improve our understanding of icebergs. If we had to include the edges for all the icebergs, we may have masked the inner behaviour. Nevertheless, when the icebergs are small, excluding the edge is simply not an option. This is also true for detection studies in which iceberg edges could be critical to identify icebergs [30].

This analysis shows an important lesson. Entropy cannot be used on its own to detect icebergs, regardless of the behaviour of the background (which may also have high entropy). For instance, one may think that icebergs are characterised by high entropy. However, if there is a dominant scatterer and we use larger windows as for Blosseville Coast N or Savissivik, the entropy may be quite low. Entropy alone will not be effective and extra information needs to be considered.

The proximity and distribution of dominant scatterers is a physical parameter for icebergs. In the future, we will investigate how we could use this for classifying different typologies of icebergs.

The difference in temperature and possibly of wet conditions seems to not affect the entropy significantly. The reason may be that there was not much liquid water during acquisition. However, it seems that for Blosseville Coast N and Savissivik (the coldest images), the span is generally among the highest and the entropy is reducing when increasing the window size. However, the latter can just

be due to the edge effect. We could therefore conclude that surface liquid water is not largely present here or not impacting much.

## 4.2. Target Characteristics

The total backscattering varies greatly going down to values around −28 dB. This is especially visible in Isortoq. In actual fact, the icebergs in Isortoq were visible for a short time, where they were floating in an area of very low sea backscattering. This corroborates the fact that open water backscattering signals may be stronger than backscatter signals from smaller icebergs [16]. Normally, these icebergs would be covered by the clutter background. Besides Isortoq, most of the icebergs are above −10 dB, showing a relatively strong backscattering in L-band. In terms of topography, higher backscatter signals could indicate smoother icebergs, and therefore, less volume scattering, which is supported by Viehoff [29].

When the entropy goes higher, this forces the average alpha to increase towards 60°. If there is a dominant mechanism in an iceberg, this seems to be a mix of surface or dipole scattering. Dipole scattering can be generated by the ice body, and therefore, this suggests volume scattering. However, in a few icebergs there is a dominant double bounce contribution, which seems to be the exception more than the norm. Additionally, the edge effect for Blosseville Coast S seems to increase the value of alpha, but it does not bring it to anything closer to dihedral scattering. In its nature, it still seems to be mostly surface scattering (probably due to surface in layover) mix to a dihedral component, which can produce something that resembles a dipole.

The values for the 5 × 5 window are mostly spread. This is most likely due to the fact that anisotropy requires a large average in order to be estimated properly. The results obtained in Figure 17 are therefore not to be taken as very significant. Once the window gets larger, we can start seeing some patterns. For instance, in Blosseville, the icebergs north and south of the glaciers seem to have different values of anisotropy. This is possibly due to different temperature effects with some presence of surface liquid water. Although, it is unlikely that icebergs in Blosseville Coast N are very wet because looking at the entropy and span, this seems to be a minor difference. Interestingly, this minor difference is visible when going to observe the two less dominant scattering mechanisms, by using the anisotropy. Specifically, the colder condition in Blosseville Coast S produces a higher entropy, which means that the second scattering mechanism is much stronger than the third. This is in line with the idea that the radiation is having a bigger penetration in the ice body and the surface scattering is accompanied by some dipole/volume scattering within the iceberg. More data will be needed on other glaciers to prove this idea is correct.

The alpha angle varies significantly for different icebergs going mostly from surface to dipoles. This suggests that icebergs can appear in images with a polarimetric behaviour which will resemble mostly a surface or volume scattering. Interestingly, the alpha angle seems to be correlated with the locations and possibly the iceberg geometry. Also of interest, we can observe differences between icebergs in Blosseville Coast N and S, although they are mostly calved by the same glacier. Again, the fact that Blosseville Coast S (the colder image) has an entropy higher than Blosseville Coast N is an indicator that we may have a larger penetration in the ice body due to less presence of surface liquid water [31].

## 4.3. Model Based Analysis

The Yamaguchi decomposition allows us to evaluate the polarimetric scattering of icebergs compared to a model considering surface, volume (of randomly oriented dipoles), double bounce and helix. Some caution must be taken when analysing these results. The volume component of the Yamaguchi model was not designed for ice, and therefore, the four basic scatterers may not be the most appropriate to analyse ice bodies. If a model is not perfectly fit to the scattering behaviour, it means that the true targets on the ground will project on the theoretical targets of the model introducing misinterpretation. In practice, this tells us that trends that we see cannot be trusted blindly but need to be interpreted in light of what other observables tell.

On first glance, in the volume vs. surface plot (Figure 22), when the backscattering signal is high, surface scattering seems to be dominant. Volume scattering becomes more dominant when the backscattering gets lower, maybe due to an increased penetration which will result in a reduction in surface scattering and an increased loss in the iceberg body. This also indicates the presence of features in the ice (cracks, crevasses, air bubbles, impurities and other features within the ice body) [22,24]. These results may indicate that the volume scattering, although present, may not be the dominant mechanism for these icebergs at L-band. In the surface vs. double bounce plot (Figure 24), it appears that surface scattering is again dominant in most of the icebergs with 5 × 5, except a few exceptions where the double bounce is stronger. Double bounce seems to be dominant only in a limited number of icebergs. However, when we compare double bounce to volume (Figure 23), the latter seems to be stronger in most cases. This shows that the pure double bounce reflection is again uncommon, and icebergs tend to have either surface or volume scattering, or a combination between surface and multiple reflections.

Our results showed a very minimal effect of helix scattering for all icebergs. In all the plots, helix seems to be the lowest scatterers. We can therefore conclude that helixes are not the most appropriate targets for observing the type of icebergs present in this scene.

Finally, regarding eventual effects of surface liquid water, observing the volume vs. surface plot (Figure 22) for Blosseville Coast N and S, we can again notice that the percentage of surface against volume is reducing when the conditions are colder. That is to say, the Blosseville Coast S points are closer to the line than the one for the N. This corroborates an increased ice penetration when the conditions are colder.

### 4.4. Summary

The general results show that volume scattering seems to be more important when the backscattering is reducing. However, the plots tell us that the most dominant scattering mechanism is surface scattering, with the second dominant being volume. This is true for icebergs which show a higher backscattering signal. These Yamaguchi results seem to corroborate with the Cloude–Pottier results.

We also investigate trends of polarimetric behaviour with temperature and eventually surface liquid water. We could compare icebergs in two close areas (Blosseville N and S), one taken in August (N) and one in June (S). Temperatures were likely to be both sub-zero. However, the solar radiation may have produced some surface liquid water on icebergs in Blosseville Coast N. Interestingly, we notice that icebergs in Blosseville Coast S (colder) generally present a higher alpha (mixture of surface and dipole scattering) and a higher anisotropy (indicating there is a second strong scattering mechanism). Additionally, the Yamaguchi decomposition also returned that the amount of volume scattering is more present in the colder image (Blosseville Coast S). We believe this may be linked to less surface liquid water, which produces a larger penetration in the ice body showing a larger volume scattering. To be finally proven, this would require a much larger dataset with more locations.

### 5. Conclusions and Further Work

Icebergs in the Arctic are subject to various backscattering behaviour during SAR image acquisition. This behaviour is dependent on iceberg properties, such as shape and size, as well as environmental and meteorological factors such as surface wetness, surface roughness and cracks/crevasses. In this study, we used five ALOS-2/PALSAR-2 L-Band SAR images to extract polarimetric information based on the backscattering behaviour of icebergs. We applied the Cloude–Pottier eigenvalue/eigenvector decomposition and Yamaguchi model decomposition by processing the average pixel values of each image. We also performed a multi-scale analysis of the images using 5 × 5 and 11 × 11 window sizes to determine the differences in polarimetric scattering behaviour and to understand if icebergs can be approximated as single targets, partial targets or a combination of single targets. In the wider scope, it is important to produce accurate iceberg detectors to ensure the enhanced safety of maritime activities, particularly due to climate change. Therefore, by performing this analysis, we also examined

the potential for icebergs to be classified based on polarimetric behaviour and geographical location. Our results show that icebergs exhibit a mix of all three polarimetric targets, but with predominance of surface and volume scattering. In some instances, double bounce can dominate the scattering, but this is rather rare. An important finding from this work is that entropy alone will not be sufficient for iceberg classification from SAR imagery.

Many icebergs exhibit a variety of characteristics depending on their locations (i.e., the glacier from which they were calved). Additionally, we could observe differences between icebergs in similar locations but at different times of the year. This suggests that colder conditions may produce more volume scattering. The analysis shows that polarimetry at L-band has potential for classifying iceberg geometry and presence of liquid water. However, to attempt classification, we would need a validation dataset where images of each iceberg structure are present, and this is currently not available.

Further work within this field includes a comparative analysis showing more iceberg locations and times of the year in the Arctic. Additionally, investigating the link between shape of icebergs via other reference data and applying PolSAR scattering models developed for glaciers. We will also investigate and compare different PolSAR detectors to identify the methodologies that seem to be more suited for detection. We also recommend estimation of the anisotropy using a large average.

**Author Contributions:** Conceptualization, J.B.; methodology, J.B. and A.M.; software, J.B.; validation, J.B. and A.M.; formal analysis, J.B.; investigation, J.B.; resources, A.M.; data curation, J.B.; writing—original draft preparation, J.B.; writing—review and editing, J.B. and A.M.; visualization, J.B.; supervision, A.M.; project administration, J.B. All authors have read and agreed to the published version of the manuscript.

**Funding:** This research received no external funding.

**Acknowledgments:** The data were provided by the project number 1151. ALOS-2 Product-JAXA 2017, all rights reserved.

**Conflicts of Interest:** The authors declare no conflict of interest. The funders had no role in the design of the study; in the collection, analyses, or interpretation of data; in the writing of the manuscript, or in the decision to publish the results.

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
