# Peer review of "Quad-Polarimetric Multi-Scale Analysis of Icebergs in ALOS-2 SAR Data: A Comparison between Icebergs in West and East Greenland"

_remotesensing, doi:10.3390/rs12111864_

Round 1

Reviewer 1 Report

Dear editor,

this study provides interesting insights into iceberg analysis from SAR imagery. Five Greenland region ALOS L-band images are used. The Cloude-Pottier and Yamaguchi decomposition methods are applied and discussed.

Although some sections seem to quickly report the analyses realized by the authors, at the end of my read I am convinced that information provided through this study is of large interest for the remote sensing polar community. New insights, issues to solve, and future perspectives are included, so representing a useful steps towards a much more complete solution for the scientific question. I realized that the authors are conscious of providing only some pieces of the puzzle (they clearly state which their aims are). This is appreciable.  

In general, the manuscript is well written and sections are complete and appropriate. Introduction is complete and provides a valid background about iceberg monitoring through SAR imagery techniques. Material and methods are very clear and well organized, so they represent a fine reference for future studies. Results are interesting and well supported by the observed outcomes. Figures are interesting but could be somewhere re-organized to help the reader follow the authors’ reasoning. Discussion is well focused; limits, approximations and uncertainties are listed and discussed objectively. This is very useful for encouraging future studies.

Hence I recommend publication after the authors have addressed the following minor suggestions:

Lines 30, 74, 242: Please check guidelines for MDPI publications about citations through the text.

Lines 34-36 as well as line 125 and following: Please, see also “The Physics of Icebergs”  by Peter Wadhams, Butterworth-Heinemann Ltd 2006.

Line 70: Check typos

Line 76: Missing space

Lines 132-133: Please note that interesting studies have been also carried out using Italian Space Agency Cosmo Sky-Med X-band imagery, as reported by Nunziata et al., 2018; Parmiggiani et al., 2018, etc. See as examples:

Nunziata et al., 2018 - Multi-Frequency and Multi-Polarization Synthetic Aperture Radar for the Larsen-C A-68 Iceberg Monitoring,  IEEE 4th International Forum on Research and Technologies for Society and Industry, RTSI 2018

Parmiggiani et al., 2018 - SAR analysis of the Larsen-C A-68 iceberg displacements, International Journal of Remote Sensing, Volume 39, Issue 18

Figure 1: Sasvissivik location cannot be identified on the map. On the other hand, I can see Kangaatsiaq in the NW but it is not mentioned into the manuscript (and located in the SW of Greenland)

Section 3.2: Please improve this section. I cannot see the point here. At lines 698-699 this information is discussed but I cannot follow the authors reasoning there looking back to this section. In general, I sincerely expected much more about the relationship with surface melting and temperature measurements (the analysis is actually limited to Tables 1 and 2).

Figure 3: Figures analysis is quite difficult due to the number and the organization of the presented images. I suggest to re-organize them in order to help the reader follow the reasoning (e.g., RGB, alpha and entropy could be presented in the same figure; presenting 5x5 before 11x11 in Figure 13 but discussing them in the opposite order, etc.). Please, evaluate if any figures group could be provided as supporting information.

Lines 471-479: Please avoid repetitions (e.g., the first step is…)

Line 565: Missing dot  

Line 609: I guess you mean Figure 19

Lines 692-694: Check typos

Lines 695-696: This information is relevant. I would clearly mention also this result in the conclusions. I think that it is quite relevant that the entropy alone is not enough to accomplish the objective of iceberg classification from SAR imagery.

Author Response

Lines 30, 74, 242: Please check guidelines for MDPI publications about citations through the text – Thanks for your suggestion, we have removed the incorrect citation formatting and rephrased the text. Please see lines 34, 170 and 340.

Lines 34-36 as well as line 125 and following: Please, see also “The Physics of Icebergs” by Peter Wadhams, Butterworth-Heinemann Ltd 2006. - Thank you for the reference, we have inserted this citation in line 57

Line 70: Check typos – Thanks, we assume this refers to the dot in between the two <> brackets. This has been removed. We also defined matrix transpose and the complex conjugate. All in line 166

Line 76: Missing space - We have added this in line 172 after the citation

Lines 132-133: Please note that interesting studies have been also carried out using Italian Space Agency Cosmo Sky-Med X-band imagery, as reported by Nunziata et al., 2018; Parmiggiani et al., 2018, etc. – Thanks for adding this references, they are indeed relevant to this study and we gladly inserted these two references into the paper, and have included a sentence on this recent work. These are in line 67.

Figure 1: Sasvissivik location cannot be identified on the map. On the other hand, I can see Kangaatsiaq in the NW but it is not mentioned into the manuscript (and located in the SW of Greenland – Sorry about this, it was a typo, we changed Kaangaatsiaq as this is incorrect and should have been Savissivik. Please see the now updated Figure 2 on page 8 of 27.

Section 3.2: Please improve this section. I cannot see the point here. At lines 698-699 this information is discussed but I cannot follow the authors reasoning there looking back to this section. In general, I sincerely expected much more about the relationship with surface melting and temperature measurements (the analysis is actually limited to Tables 1 and 2). – Thanks for the suggestion, we have added two extra paragraphs describing the relationship with surface melting and temperature measurements, and we have also noted that the midnight sun has an effect on these temperatures, as well as geographical location. Please see the section 2.3, 2.4 and 2.5 on pages 8-10.

Figure 3: Figures analysis is quite difficult due to the number and the organization of the presented images. I suggest to re-organize them in order to help the reader follow the reasoning (e.g., RGB, alpha and entropy could be presented in the same figure; presenting 5x5 before 11x11 in Figure 13 but discussing them in the opposite order, etc.). Please, evaluate if any figures group could be provided as supporting information. – This is a good point and we agree we were showing too many and too big figures. We have fully revised all figures. Now we have a combination of figures that show different parameters packed together. Also we are not showing everything of each location. We have added a scale, a north arrow, and coordinates in DMS to address the cartographic issues, as well as a colourmap to indicate alpha and entropy values.

Lines 471-479: Please avoid repetitions (e.g., the first step is…) – This sentence has been deleted, as this is not adding much to the following analysis. Please see lines 384 onwards for the changes.

Line 565: Missing dot - Revision added, line 498.

Line 609: I guess you mean Figure 19 - Yes, we have revised this. This is now Figure 14.

Lines 692-694: Check typos - Thank you for pointing this out, we have corrected the spelling of 'collection', indicated a), b) and c) for each location respectively, and corrected 'reduced' to 'reduce'. Line 569-578.

Lines 695-696: This information is relevant. I would clearly mention also this result in the conclusions. I think that it is quite relevant that the entropy alone is not enough to accomplish the objective of iceberg classification from SAR imagery. – Thanks for the suggestion, we have stated this as one of the main findings within the conclusions, in line 726-727.

Reviewer 2 Report

Clear and well written paper.

Author Response

Thank you very much for reviewing the manuscript. We hope the revised edition will be an improvement. We have checked the English spelling prior to resubmission.

Reviewer 3 Report

The contribution evaluates five ALOS-2 quad-pol acquisitions for the characterization of icebergs. PolSAR Decomposition features are compared for different locations in the Arctic. The investigated icebergs were visually identified from the SAR images.

The topic and the general approach are of interest, while merit and value of the conducted analyses needs to be pointed out in a better way. It is somehow unclear how the findings advance the understanding of SAR scattering from icebergs. I might get clearer to the reader once the revisions are done.

Anyway, the manuscript in its present form is not suited for publication and major revisions are required:

# Figures do not follow the requirements specified by the journal, i.e. all cartographic elements beside the image itself are missing. Check the guidelines how to prepare the figures carefully, i.e. section “Preparing Figures, Schemes and Tables”. Besides that, Figures 2-11 and Figure 13-24 are too big/large and unsuited for publication in the present form. More information on the imagery, location ect. must be added to the figure captions as well.

# References and Bibliography do not follow the requirements specified by the journal. Check the guidelines, section “Manuscript Preparation”. Carefully check prior to the resubmission.

# The structure of the manuscript needs major revisions, e.g. First part of “3. Results” should go to section “2. Material and Methods”. Information that is right now provided in section 2. should be moved to “1. Introduction”. Section 3 mixes results and discussion. Again, check the provided guidelines, section “Research Manuscript Sections” there is clear information on what to include in the sections.

# Methodology needs a more complete description. Information on the processing of the SAR data and on the detection of Icebergs must be provided. The results on the visually identified Icebergs are not provided; these need to be shown in comparison to the images. It is unclear how analysis was finally conducted and why.

# The discussion is not complete; the authors should discuss the results and how they can be interpreted in perspective of previous studies and of the working hypotheses.

# Formatting of the manuscript needs major revision, e.g. Figure Captions are incomplete and do not give the relevant information. Figures and Captions are placed on different pages, images are too big, legend is too small, some sections are just a single sentence, ect.

I did like to point out that a careful preparation of a manuscript is mandatory. Reviews take a lot of (unpaid) time and do even take more time if guidelines provided are not read or simply ignored.

Kind regards

###########################################

L12/L13: in west and east Greenland (?)

L15 ff.: “… show that the main scattering components … surface, double bunce and volume scattering.” That is too generic, as this mostly holds for all targets.

L18: “using different window sizes” more information required to understand, what window sizes?

L30: space before Fettweis missing

L35: “Their size … “ that information accounts for the Arctic ones?

L38/39: “… using a physical detector or decomposition.” Unclear what that means? Does PolSAR analysis enhances the general ability to indicate icebergs? Or is the main propose of the PolSAR data to get more characteristics of the Icebergs?

L40: First sentence, more background required, what is “multiple observables”, i.e. multiple SAR images?

L42: “some polarimetric operator as the Span.” You mean general operators that can be deduced independent of a decomposition? What’s with the “pure” amplitudes/intensities (S2)?

L45 ff.: Not needed in my opinion (i.e. cancel), the paper is short enough to keep the overview

L53: Check reference for equations in text using the template.

L68: so incoherent target decompositions

L74: Missing reference on Chandrasekhar

L76: Must be filtered, I will never be removed completely

L77: Missing blank after the references

L82: Application to the C3 is also done, results in same entropy but different angles compared to T3. Maybe not of relevance here though.

L83: Pauli basis not defined.

L95: Most commonly displayed from 0° to 90°, isn’t it?

L107: Reference to these studies missing, make a short selection?

L115 ff.: As this is an incomplete list of equations, also not introducing the different cases, I suggest removing these equations. They are not required to understand you approach and the decomposition is known to the community quite well. Same goes for the Cloude-Pottier decomposition, but it is up to you to keep it or not. Your description and some references might do the job.

L79 ff.: I suggest moving Table 4 to this section and include a reference in the text? General question: the Yamaguchi decomposition features and the span were converted to dB prior to the analysis?

L119 ff.: Information of these sections can just go in the introduction? It is the state of the art on iceberg detection with SAR that is provided here, so best place would be the introduction. You should come back to some of the references later in the discussion.

L132: rephrase sentence “carry” sounds odd to me?

L133: 30 m feasible, but might be better or coarser depending on the used acquisition mode?

L160: cancel or rephrase first sentence

L160 ff.: Include a sentence on why PolSAR is beneficial for detection in general.

L169: Extracting the 3D; what 3D? i.e. what is meant here? Not clear to the reader

L170: One study; name the authors and then include reference [19]; check journal’s template

L180: papers; these studies

L170-182: This section needs a revision. There is no clear line of arguments, it reads like a list of references with some notes.

L184: I wouldn’t’ call it unpredictable, we do have weather forecasts, also for the Arctic. As well, Arctic is not defined. Cancel or rephrase sentence.

L184: Constantly changing; not true. There might be stable weather for two weeks.

L185: Cancel or rephrase sentence. That sounds colloquial.

L187: One study; name authors, check journal’s template for information

L209-226: Move this information to the section on Materials and methods. It is description of the study area and not part of the results.

L119: Figure 1: see general comment at the beginning and revise figure accordingly.

L272 – 240: Move this information to the section on Materials and methods. It is description of the study area and not part of the results.

L232: Table 1: It is averaged data from one day? Or average according to the SAR image acquisition time? That is not clear, provide more information in the Table caption and in the main text.

L233: That is local time? So images were taken 1-3 hours after “midnightsun”? What is with the lightning conditions?

L242: Check journal’s template on the reference.

L244 ff.: So every iceberg was assigned to one glacier? How reliable is this assignment? Are the icebergs close to the respective glacier?

L241-L250: Move this information to the section on Materials and methods. It is description of the study area and not part of the results.

L251-L508: Move this information to the section on Materials and methods. It is description of the used data and not part of the results. The section needs revision. Figures: see general comment at the beginning and revise figures accordingly. Figures can be separated from the data description and moved to the results section.

L251- 267: More information on the SAR data and their processing needed. List and describe all processing steps, including speckle filtering, multi-looking, decomposition estimation windows, georeferenceing, ect. Your approach is not clear from the provided description. Was analysis done on the slant-range imagery, if yes, why? Was a DEM used in the processing? What processing level (beta, sigma, gamma) was used? Were Entropy/Alpha used to detect Icebergs, or is visual detection based on Span only, or Pauli RGB Composite?

Table 3: Lat/Lon: That is the center coordinate?

L477: Information on beta not provided in the methods section

L479: The relation can be quantified by calculating a correlation coefficient?

Figure 12: Too small, more information in the caption required.

L489 ff.: avoid meta language

L509 ff.: This section mixes results, interpretation and discussion. Shorten condense and provide objective description. Move other text to discussion.

L676 ff.: The discussion misses a placement of your findings regarding related literature. There is not a single reference in this section.

L698: That might not be true for the scenes acquired in June/August as we might have polar days? So I would not exclude the influence of liquid water. That needs more discussion. Also see your own statement in L707.

L748: What is with the mentioned models for scattering from ice, or investigating the actual shape of Icebergs via other reference data? These points seem more logic to me then a comparison to Antarctic icebergs?

Author Response

# Figures do not follow the requirements specified by the journal, i.e. all cartographic elements beside the image itself are missing. Check the guidelines how to prepare the figures carefully, i.e. section “Preparing Figures, Schemes and Tables”. Besides that, Figures 2-11 and Figure 13-24 are too big/large and unsuited for publication in the present form. More information on the imagery, location ect. must be added to the figure captions as well. – Sorry about missing this, we should have checked it out. We amended most of the figures, we added all cartographic elements to the images in Figures 2 to 7. The plots in Figures 8 to 18 are now presented such that the 5x5 and 11x11 plots are in one figure. Information has been added to the figure captions.

# References and Bibliography do not follow the requirements specified by the journal. Check the guidelines, section “Manuscript Preparation”. Carefully check prior to the resubmission. – Thanks for spotting this, we have revised the manuscript and the MDPI reference format has been applied to the bibliography using EndNote, and citations have been addressed. Thanks again for spotting inconsistency with this and we hope that now it is all following the journal format

# The structure of the manuscript needs major revisions, e.g. First part of “3. Results” should go to section “2. Material and Methods”. Information that is right now provided in section 2. should be moved to “1. Introduction”. Section 3 mixes results and discussion. Again, check the provided guidelines, section “Research Manuscript Sections” there is clear information on what to include in the sections. – We agree the manuscript needed some restructuring. The first part of the results has been moved to the methods (section 2.3). Information originally in the methods is now in the introduction (section 1.1, 1.1.1, 1.1.2, 1.1.3, and 1.1.4) and information in the results section has been moved to the discussion (section 4.1, 4.2, and 4.3).

# Methodology needs a more complete description. Information on the processing of the SAR data and on the detection of Icebergs must be provided. The results on the visually identified Icebergs are not provided; these need to be shown in comparison to the images. It is unclear how analysis was finally conducted and why. – Thanks for the suggestion. To make clearer our processing stack, we included a block diagram in Figure 1. Information on the processing of SAR data and the detection of the icebergs is now provided. One of the new figures (Figure 3) also shows a closer look at icebergs in the images. We also added much more info on how we identified icebergs in images.

# The discussion is not complete; the authors should discuss the results and how they can be interpreted in perspective of previous studies and of the working hypotheses. – Thanks for the suggestion, the discussion section has been updated with more concepts and with references. I hope it is fine, but please let us know if there is some more to modify/add there.

# Formatting of the manuscript needs major revision, e.g. Figure Captions are incomplete and do not give the relevant information. Figures and Captions are placed on different pages, images are too big, legend is too small, some sections are just a single sentence, ect. – Thanks and sorry about this, we tried to review the full formatting of the manuscript.

I did like to point out that a careful preparation of a manuscript is mandatory. Reviews take a lot of (unpaid) time and do even take more time if guidelines provided are not read or simply ignored.

Kind regards

###########################################

L12/L13: in west and east Greenland (?) – This has been specified in line 13

L15 ff.: “… show that the main scattering components … surface, double bunce and volume scattering.” That is too generic, as this mostly holds for all targets. – We understand this reviewer point, indeed we were too generic with this. The abstract is now updated to show a better summary of the results. See lines 15-18.

L18: “using different window sizes” more information required to understand, what window sizes? We made this more specific, saying boxcar 5x5 and 11x11 window sizes, information now in line 20-22

L30: space before Fettweis missing – This line has been updated, please see line 34

L35: “Their size … “ that information accounts for the Arctic ones? – This information is now in line 39

L38/39: “… using a physical detector or decomposition.” Unclear what that means? Does PolSAR analysis enhances the general ability to indicate icebergs? Or is the main propose of the PolSAR data to get more characteristics of the Icebergs? – We agree that the sentence was confusing. We have modified the text and indicated that PolSAR analysis will be useful for detection and classification. The information is in line 42 onwards.

L40: First sentence, more background required, what is “multiple observables”, i.e. multiple SAR images? – We updated this starting from line 42.

L42: “some polarimetric operator as the Span.” You mean general operators that can be deduced independent of a decomposition? What’s with the “pure” amplitudes/intensities (S2)? – Sorry for the confusion expression, we have updated this as, “an overall intensity operator (span)” now in lines 48.

L45 ff.: Not needed in my opinion (i.e. cancel), the paper is short enough to keep the overview – Thanks for the suggestion, this part has been removed.

L53: Check reference for equations in text using the template. – Thanks for this suggestion, we have revised the full formatting of the paper and we hope it is now following the journal suggested format. Specifically, equations should be in the text, and bracket numbers in the text removed to avoid confusion with square brackets for citations.

L68: so incoherent target decompositions – Thanks, information now in line 174

L74: Missing reference on Chandrasekhar – We added the reference, line 170

L76: Must be filtered, I will never be removed completely - That is correct we rephrased this in line 172.

L77: Missing blank after the references – space added, line 172.

L82: Application to the C3 is also done, results in same entropy but different angles compared to T3. Maybe not of relevance here though. – Thanks for the suggestion. Here we prefer to use the T3 matrix so that we can easily interpret the alpha angle using the ordinary Cloude-Pottier formalism.

L83: Pauli basis not defined. – Thanks, we defined the Pauli basis now in lines 203-205.

L95: Most commonly displayed from 0° to 90°, isn’t it? – This is correct we changed this in line 198

L107: Reference to these studies missing, make a short selection? – References to the studies are now added, lines 222-223.

L115 ff.: As this is an incomplete list of equations, also not introducing the different cases, I suggest removing these equations. They are not required to understand you approach and the decomposition is known to the community quite well. Same goes for the Cloude-Pottier decomposition, but it is up to you to keep it or not. Your description and some references might do the job. – We agree, it is right, they are not really needed here. We have removed the Yamaguchi equations, but we have decided to keep the Cloude Pottier ones.

L79 ff.: I suggest moving Table 4 to this section and include a reference in the text? General question: the Yamaguchi decomposition features and the span were converted to dB prior to the analysis? – Table 4 is now on page 7 of 27. Yamaguchi and span were converted to dB as the very last step of the analysis. We added some info on this in a block diagram.

L119 ff.: Information of these sections can just go in the introduction? It is the state of the art on iceberg detection with SAR that is provided here, so best place would be the introduction. You should come back to some of the references later in the discussion. – Thanks for the suggestion; this section is now in the introduction [section 1.1, 1.1.1, 1.1.2, 1.1.3 and 1.1.4]

L132: rephrase sentence “carry” sounds odd to me? – We modified it into ‘have’ at line 64.

L133: 30 m feasible, but might be better or coarser depending on the used acquisition mode? – We added that this varies depending on the acquisition mode, in line 65.

L160: cancel or rephrase first sentence – This change is now in lines 92-93.

L160 ff.: Include a sentence on why PolSAR is beneficial for detection in general. – We added that PolSAR helps identify multiple scattering mechanisms and avoids focusing on a fixed scattering mechanism where the iceberg may be very weak or the background may be very strong. This sentence is now in lines 93-95.

L169: Extracting the 3D; what 3D? i.e. what is meant here? Not clear to the reader – What we meant was iceberg topography. We have changed the sub heading into iceberg topography – line 103.

L170: One study; name the authors and then include reference [19]; check journal’s template – This has been corrected in line 106.

L180: papers; these studies – correction in line 116.

L170-182: This section needs a revision. There is no clear line of arguments, it reads like a list of references with some notes. – Thanks for the suggestion. This section has now been updated and we tried to show our argument better, lines 104-120.

L184: I wouldn’t’ call it unpredictable, we do have weather forecasts, also for the Arctic. As well, Arctic is not defined. Cancel or rephrase sentence. – The sentence was rephrased to “In Greenland, snow is prone to melting, and rainwater may fall from clouds within areas of warmer water where the iceberg may drift” in line 122.

L184: Constantly changing; not true. There might be stable weather for two weeks. – Sorry for this superficial statement, we just meant that the weather does not stay stable on very long time frames. This part is removed.

L185: Cancel or rephrase sentence. That sounds colloquial. – Sorry about this, the sentence in line 122 should have also addressed this.

L187: One study; name authors, check journal’s template for information – information added in line 125.

L209-226: Move this information to the section on Materials and methods. It is description of the study area and not part of the results. – Thanks for the suggestion, we did this and have moved the information to section 2.3

L119: Figure 1: see general comment at the beginning and revise figure accordingly. – We indeed revised most of the figures. We hope that now they are conforming to the journal standard.

L272 – 240: Move this information to the section on Materials and methods. It is description of the study area and not part of the results. – This information has been moved, section 2.4

L232: Table 1: It is averaged data from one day? Or average according to the SAR image acquisition time? That is not clear, provide more information in the Table caption and in the main text. – This was the monthly average, unfortunately we could not get daily data. We updated table 2 with this information.

L233: That is local time? So images were taken 1-3 hours after “midnightsun”? What is with the lightning conditions? – This is correct, we have added several information regarding the effects of sunlight on eventual surface liquid water, see lines 332-338.

L242: Check journal’s template on the reference. – We corrected this and now is at line 340.

L244 ff.: So every iceberg was assigned to one glacier? How reliable is this assignment? Are the icebergs close to the respective glacier? – The icebergs were tens of km from the glacier terminus however we also added in the paper that drift may have occurred. Line 349-351

L241-L250: Move this information to the section on Materials and methods. It is description of the study area and not part of the results.  – This information has been moved, section 2.5

L251-L508: Move this information to the section on Materials and methods. It is description of the used data and not part of the results. The section needs revision. Figures: see general comment at the beginning and revise figures accordingly. Figures can be separated from the data description and moved to the results section. – This information has been moved, section 2.6. The figures are updated and in the results section.

L251- 267: More information on the SAR data and their processing needed. List and describe all processing steps, including speckle filtering, multi-looking, decomposition estimation windows, georeferenceing, ect. Your approach is not clear from the provided description. Was analysis done on the slant-range imagery, if yes, why? Was a DEM used in the processing? What processing level (beta, sigma, gamma) was used? Were Entropy/Alpha used to detect Icebergs, or is visual detection based on Span only, or Pauli RGB Composite? – Sorry for not adding more info on this. We have included a block diagram to describe the processing steps at line 234. We used slant range imagery because when geo-referencing there is the risk of losing polarimetric information. The common guideline in this is to do all the quad-pol processing in SLC and geocode only the output mask. Since we have no output masks (but plots) here we did not need to geocode and we preferred to keep the images all in radar coordinates. This is just to make sure we are not biasing the quad pol results in some ways. We used the ALOS-2 ancillary data to find out what the coordinates of each pixels are for visualisation purposes.

Table 3: Lat/Lon: That is the center coordinate? – Yes, now specified in the caption.

L477: Information on beta not provided in the methods section – Thanks for spotting this, we have now added information on beta at line 198-199.

L479: The relation can be quantified by calculating a correlation coefficient? – This is true, however we have decided collinearity is not present and the analysis is not really relevant for understanding the iceberg polarimetric behaviour. Since we have already lots of figures we decided that this is as unneccesary here and have removed this part.

Figure 12: Too small, more information in the caption required. – As above, this figure was removed.

L489 ff.: avoid meta language – update in line 443-445

L509 ff.: This section mixes results, interpretation and discussion. Shorten condense and provide objective description. Move other text to discussion. – Thanks for this suggestion, we moved this information on interpretation to the discussion section.

L676 ff.: The discussion misses a placement of your findings regarding related literature. There is not a single reference in this section. – The discussion section now provides an interpretation of the results in context with the literature, with some references now added.

L698: That might not be true for the scenes acquired in June/August as we might have polar days? So I would not exclude the influence of liquid water. That needs more discussion. Also see your own statement in L707. – This information has now been updated. Lines 688-692

L748: What is with the mentioned models for scattering from ice, or investigating the actual shape of Icebergs via other reference data? These points seem more logic to me then a comparison to Antarctic icebergs? – This is a very good point, thanks for the suggestion. We changed the future work into: “Further work within this field could include a comparative analysis including more iceberg locations and times of the year in the Arctic. Additionally, investigating the link between shape of icebergs via other reference data and applying PolSAR scattering models developed for glaciers.” at lines 738-742.

Reviewer 4 Report

Dear representatives of the Remote Sensing editorial board
and authors of the manuscript # 794400,

The topic is interesting and useful for SAR based iceberg identification
and classification. However, the manuscript will still needs to improved
before being able to published.

General comments:

1) The structure of the manuscript needs to be updated.

Subsection 2.2. in my opinion this could be part of the introduction (or
in a separate background section after introduction).

Sections 3.1-3.4 should be in a data section, which should be after the
introduction (and possible background section). The section 3.4. should
be first in the data section as SAR data are the most important data in this
study.

Sections 2.1. forms the Methods section.

The sections starting from 3.5 on are results, so they are in their correct place.

The structure would then be
Introduction
(Background)
Data and study area
Methodology
Results
Discussion
Conclusions

2) It is not fully clear which of the icebergs are in open water and which within
sea ice. Would it be possible to draw ice edge e.g. in the RGB images.
Also give the numbers of icebergs in open water and in sea ice.

3) What bothers me is the identification of icebergs made visually and without
any supporting data. How well icebergs are really distinguished e.g. from
ice floes of the same magnitude in area?
Would it be possible to include some supporting data e.g. from high-resolution
optical/IR/NIR instruments? Or at least include some kind of comments on how
successful the identification step could be.

4) It would also be interesting to see a comparison of the visual iceberg
identification using only single-pol, dual-pol and quad-pol. How much does
adding polarization channels improve the visual detection. Also comparisons
to simple single-pol and dual-pol features computed for detected icebergs
would be interesting, such features could be e.g. the size of the IB in pixels,
contrast w.r.t background or SNR etc. It would give some idea on how much
adding of polarization channels will improve the classification of icebergs
now that the information on the scattering mechanisms can be retrieved from
the quad-pol data.

5) An interesting question is detection of icebergs: what would be the
benefits of using quad-pol information and how much could the quad-pol
decompositions possibly improve detection. Have You any comments to be included
on how e.g. the extracted scattering mechanisms could be used to improve
iceberg distinguishing from cluttered open water signal or from ice floes?
I think distinguishing between ships and iceberg using high-resolution
quad-pol SAR could also be possible. Of course, these are already topics
of another manuscript but it would be good to see some ideas how the
work performed here could be utilized to improve detection.

Some detailed comments:

Open ALOS-2 and PALSAR-2 when they appear for the first time.
ALOS-2 is the platform and PALSAR-2 is the instrument, it is better to say
"ALOS-2/PALSAR-2 SAR data" (e.g. in the abstract) and use ALOS-2/PALSAR-2 throughout
the manuscript.
P7 L252: "ALOS-2 PALSAR-2 JAXA radar satellite": rather say "PALSAR-2 instrument
aboard ALOS-2 satellite"
P7 L252: Open JAXA
P8 L267: "Pauli RGB processed image", indicate also which parameter is on each
Pauli image channel and how they are scaled.

Sincerely,

LocalWords: SAR RGB NIR IB SNR decompositions ALOS PALSAR JAXA

Author Response

1) The structure of the manuscript needs to be updated.

Subsection 2.2. in my opinion this could be part of the introduction (or

in a separate background section after introduction).

Sections 3.1-3.4 should be in a data section, which should be after the

introduction (and possible background section). The section 3.4. should

be first in the data section as SAR data are the most important data in this

study.

Sections 2.1. forms the Methods section.

The sections starting from 3.5 on are results, so they are in their correct place.

The structure would then be

Introduction

(Background)

Data and study area

Methodology

Results

Discussion

Conclusions

  • Thanks for the help with this, we are sorry for the confusion with the previous structure. Other reviewers have also pointed out we need to change the structure and we have done this. Trying to combine suggestions from different reviewers we did not include a named Background section and most of this has now been moved to the introduction, sections 1.1, 1.1.1, 1.1.2, 1.1.3 and 1.1.4. We have included the data and study area in the Material and Methods section, in sub sections.

2) It is not fully clear which of the icebergs are in open water and which within

sea ice. Would it be possible to draw ice edge e.g. in the RGB images.

Also give the numbers of icebergs in open water and in sea ice.

  • This is an interesting point and in the future we will try to investigate this using reference data. From what we can see it seems that we have mostly sea ice, although in some areas the background backscattering is so low that we cannot really say if it is open ocean or it is very thin ice. I would tend to see, very calm open ocean. In other areas there is a large mix of floes floating in open ocean areas. We have added a sentence in the caption of images to try to depict what is the main background for that image. We prefer to not draw sea ice line because in some images we have a very fragmented scenario and these extra info may be confusion.

3) What bothers me is the identification of icebergs made visually and without

any supporting data. How well icebergs are really distinguished e.g. from

ice floes of the same magnitude in area?

Would it be possible to include some supporting data e.g. from high-resolution

optical/IR/NIR instruments? Or at least include some kind of comments on how

successful the identification step could be.

  • Unfortunately we did not find any optical image that could be used at the time of acquisitions. Some of the images were taken under cloud cover and others at different times so not comparable. We have added more information on how we identified icebergs and we also added some text in the discussion section. We tried do use a conservative way of identifying icebergs although this inevitably will be missing the more “stealth” icebergs that will not show any clear feature.

4) It would also be interesting to see a comparison of the visual iceberg

identification using only single-pol, dual-pol and quad-pol. How much does

adding polarization channels improve the visual detection. Also comparisons

to simple single-pol and dual-pol features computed for detected icebergs

would be interesting, such features could be e.g. the size of the IB in pixels,

contrast w.r.t background or SNR etc. It would give some idea on how much

adding of polarization channels will improve the classification of icebergs

now that the information on the scattering mechanisms can be retrieved from

the quad-pol data.

–These are all excellent points and they will be keeping us busy for the next months or years. However, we would prefer to leave the detection capability as the argument of a second study. This is because we don’t like to come with preliminary results about analysis of simple features but we want to compare a large set of state of the art detectors using different polarimetric modes and very different detection rationale (e.g. intensity, scale invariant parameters, sub-look analysis). We want to include the use of quantitative metrics like ROC or SCR and evaluate quantitively how much we improve in detection by using quad pol. This is a very large work and cannot be included here, also because the focus of this paper is more on characterisation than detection. Coming back on the way we did identification, we added some information on the fact that we used the Pauli RGB (and looked mostly at “image features” typical of icebergs), but we would prefer to leave the comparison of single and quad-pol data for detection as the topic for another study.

5) An interesting question is detection of icebergs: what would be the

benefits of using quad-pol information and how much could the quad-pol

decompositions possibly improve detection. Have You any comments to be included

on how e.g. the extracted scattering mechanisms could be used to improve

iceberg distinguishing from cluttered open water signal or from ice floes?

I think distinguishing between ships and iceberg using high-resolution

quad-pol SAR could also be possible. Of course, these are already topics

of another manuscript but it would be good to see some ideas how the

work performed here could be utilized to improve detection.

  • Thanks for this suggestion. We added some comments on this in the dissemination. This study is important to inform the design of a new detector as well as help separate icebergs from ships. There is lot of work on machine learning trying to do this and we believe by using better feature vectors to train the machine we would be able to improve the results.

Some detailed comments:

Open ALOS-2 and PALSAR-2 when they appear for the first time.

ALOS-2 is the platform and PALSAR-2 is the instrument, it is better to say

"ALOS-2/PALSAR-2 SAR data" (e.g. in the abstract) and use ALOS-2/PALSAR-2 throughout

the manuscript. – Thank you, these changes have now been made.

P7 L252: "ALOS-2 PALSAR-2 JAXA radar satellite": rather say "PALSAR-2 instrument

aboard ALOS-2 satellite" – This information is now in Line 360

P7 L252: Open JAXA – This is now in line 361-362

P8 L267: "Pauli RGB processed image", indicate also which parameter is on each Pauli image channel and how they are scaled – Thank you for the suggestion. For each subset image in Figures 3 to 7, we have indicated each parameter (alpha, entropy) and the window sizes. We added that the RGB contains the Pauli components in linear scale. Figures for the images now contain information on the scale in km. We hope this information is more accessible.

Round 2

Reviewer 3 Report

Dear authors,

Thanks for addressing my review comments and for answering my questions on point-to-point basis. The quality of the manuscript hast increased considerably, especially the quality of presentation. The new structure makes it now easier to follow and to understand your approach.

I do have some minor issues that need to be addressed/discussed before manuscript is ready for publication. These are listed in the following.

Kind regards

##############################

L26 ff.: Good introduction that nicely presents the state of the art with a good line of argument. As mentioned later you might give more details on the objectives of the study. Suggestion: Just cancel the “third-level” headings (e.g. 1.1.1. Intensity detectors).

L108: Reference for Herdes missing (?)

L113: Name author(s)

L116: These studies -> this study?

L142: Move your objectives to a new section and give more details on the goals of the study, maybe 2-3 sentences

L146: You may place some lines here that give a brief overview on what you did, e.g. on Greenland, ALOS data and you applied decompositions. Details are provided later, but still a short summary might be beneficial for the reader. Suggestion: You might move parts of section 2.6 to the beginning of this chapter and then continue with the description of the decompositions and the study area.

L146 ff.: Check the notation and definitions. Paulis basis (Equation 10) is now defined as ?; however, in Equation 4 same symbol (k) is used for the definition of the covariance matrix? That might be confused, as target vector (?) given in Equation 9 is the true basis for the covariance matrix? Maybe I am missing something, so please check and make consistent.

Figure 1: Thanks for the figure. Even though there are not too many processing steps that figure helps to keep the overview.

L274: the final block -> sounds odd to me

L275: What PolSAR variables were shown in the RGB; which features were chosen for the R, G and B channels?

L277/278: in the direction opposite to the sensors -> sounds odd. Maybe better: in looking direction or line-of-sight direction? Not sure, though.

L292: identify by visual analysis

Figure 2: I think the screenshot from google earth will be ok for the journal; however, I suggest reworking the figure. It would be nice, if you can link the overview image on the location of the SAR acquisitions, with zooms that show the actual footprint of the SAR data with higher level of detail, e.g. together with the coastline of Greenland. That will help to localize your study areas and will also help to understand the context, e.g. if scene is close to the coast or glaciers ect.. You may also include the location of the climate stations in this map (see L312). You might check the GIMP data for the basemap: https://nsidc.org/data/measures/gimp

L316: In your answer you pointed out that the displayed information is the monthly average: “L232: Table 1: It is averaged data from one day? Or average according to the SAR image acquisition time? That is not clear, provide more information in the Table caption and in the main text. – This was the monthly average, unfortunately we could not get daily data. We updated table 2 with this information.” So how can there be a minimum temperature if you do not have the daily temperatures, but just the monthly average? Please make clear in text and table.

L324: Important to note/calculate if this is the local time. Is it the local time or the UTC (whatever)?

L332 ff.: Thanks for adding this section.

L345: provide the period? Around 2017 sounds odd

Table 3: If GIS data on these glaciers is available you might add this in addition to the footprint and the coastline in your reworked figure 2.

L373: In my opinion you can cancel this sentence.

L359 ff.: Suggestion: As mentioned above you might move the description of the SAR data to the very beginning of the chapter. You might keep the information on the visual detection, as this then nicely reads as a continuation of what you developed in 2.5.

Figure 3: Thanks for the update of the figures. Much better now. You might indicate the extent of Figure 4 in Figure 3 a. Same goes for the other figures.

Figure 3-7: It would be nice to have an additional subfigure showing the same extent and indicating the visually identified icebergs, e.g. as dots/polygons. That would allow to compare the identified icebergs with the PolSAR features.

L415: cancel “bright” or move before “azimuth”

Figure 8 – 18: Check if the colours fit the legend. To me it seems that colours red and orange are switched when comparing the subfigures a) and b), the results from different estimation window sizes respectively. Best visible in Figure 8 a in comparison to Figure 8 b. Please check also in the other figures.

Suggestions: (i) Combine Figures 8 to 12 in one figure? That would save some space and enhance the comparison/presentation?  Same might work for Figures 13-18? (ii) For the most important features (e.g. Entropy), you might draw the boxplots across the study areas? That should nicely underline how the icebergs from different regions differ for some of the PolSAR features and for different estimation window sizes? You might include this somehow as a visual summary, as you start your discussion on this?

Figures 13-18: That is just logarithmic scale of the intensity values? So it is not the actual dB value that would require a different transformation, also for the PALSAR images? I might be wrong, but the conversion from linear to dB is not done using the 10*ALOG10(X) function for ALOS imagery, but there is a constant term that needs to be subtracted?

L493 ff.: Wrong text format.

L565: As mentioned above, at some stage of the manuscript the results from the visual analysis should be shown along the SAR features. This is of importance, as your reference data is so far not shown in the manuscript.

L574: Suggestion: Focusing on the change of the Entropy values with changing estimation window sizes: That might be nicely highlighted if you draw the above-mentioned boxplots.

L615: the window size.

L619ff.: Either include as list or remove the capital letter abbreviations. It will be clear from the text.

L619: The -20dB is not low, considering that the noise floor is to be expected at <-30 dB for L-Band? Might be related to the dB conversion?

L620: stroke of luck sounds colloquial; this fact underlines the need for the reader to actually see your results from the visual analysis, so that he or she can judge on the reliability of your results.

L623: Missing “.” and space after reference

L624: large -> strong (?)

L629: mathematically not needed in my opinion

L628 ff.: Interesting observation. In fact my first thought on this was the edge effect, but as you developed, there might be different reasons for this effect.

L639: Wrong format for cross reference (“17a”)

L641/642: Any reference that would support this statement (“presence of surface liquid water”) ?

L637: Interesting observation and in fact worth for future work.

L669 & L675: include cross-reference to figure

L678: comma after however (?) and include cross-reference to figure

L687: typo pvolume and include cross-reference to figure

If you continue your work and you look for more suited sites, you might consider the north eastern coast of Greenland. The stations Daneborg and Zackenberg do record weather information and most of the data is free and open.

Reviewer 4 Report

Dear editorial board members of RS and authors of the manuscript
emotesensing-794400.v2,

The authors have replied to the reviewer comments and taken the
comments into account where possible with a reasonable work.
After some minor polishing the manuscript will be ready for
publication.

Some minor comments, in random order:

P10, SAR dataset: Also give the incidence angle range, now only
average is given. Give the range either for each SAR image in the
table or for all the images jointly if the ranges are very similar.

Figures 3-7: It would be good to indicate land areas in the imagery.
Either apply a land mask or include coastlines in the figures (where
applicable).

P4 L145: "Materials and Methods". Now you first present the methods
and then materials. If the order is this the title should then be
"Methods and Materials". The other alternative is to move the
subsections related to the material to the beginning of the section.

P6 L315: "Meteorological Information". This is almost the same as
"meteorological data" in the previous subsection title. I think
e.g. "meteorological conditions" would be better here.

P6 Table 2. Why are the 20 June and 15 August weather data not
included in the table? If not good weather measurements are available
you could also consider using weather model reanlysis data for this
(e.g. NCEP/NCAR reanalysis should be available).

Now there are many paragraphs beginning with polarimetric feature
or scattering mechanism name (such as "ALPHA", "BETA" or "VOLUME
VS SURFACE", followed by a colon. These paragraphs should be
started by a proper sentence rather, such as "The plot diagram(s) for
alpha is/are presented in Fig. X." And if the form a list, like in
subsection 4.2 on page 22, You could start with a sentence like:
"Here we evaluate the target characteristics be polarimetric
parameters:"
And then a numbered/labeled list, e.g.:
a) Span. ...
b) Alpha. ...
c) Anisotropy. ...
